

# Regional variations in mineralogy of dust in ice cores obtained from northeastern and northwestern Greenland over the past 100 years

Naoko Nagatsuka[1], Kumiko Goto-Azuma[1, 2], Koji Fujita[3], Yuki Komuro[4], Motohiro Hirabayashi[1], Jun Ogata[1], Kaori Fukuda[1], Yoshimi Ogawa-Tsukagawa[1], Kyotaro Kitamura[1], Ayaka Yonekura[7], Fumio Nakazawa[1], Yukihiko Onuma[5], Naoyuki Kurita[6], Sune Olander Rasmussen[8], Giulia Sinnl[8], Trevor James Popp[8] and Dorthe Dahl-Jensen[8]

[1] National Institute of Polar Research, Tokyo 190-8518, Japan
[2] Department of Polar Science, The Graduate University for Advanced Studies, SOKENDAI, Tokyo 190-8518, Japan
[3] Graduate School of Environmental Studies, Nagoya University, Nagoya 464-8601, Japan
[4] Laboratory for Environmental Research at Mount Fuji, Tokyo 169-0072, Japan
[5] Earth Observation Research Center (EORC), Japan Aerospace Exploration Agency (JAXA), Tsukuba, 305-8505, Japan
[6] Institute for Space-Earth Environmental Research, Nagoya University, Nagoya 464-8601, Japan
[7] Marine Works Japan Ltd., Kanagawa 237-0063, Japan
[8] Physics of Ice Climate and Earth, Niels Bohr Institute, University of Copenhagen, Tagensvej 16, DK-2200, Copenhagen N, Denmark

**Correspondence**: Naoko Nagatsuka (nagatsuka.naoko@nipr.ac.jp)

**Abstract.**

To investigate regional and temporal variations in the sources and atmospheric transport processes for mineral dust deposited on the Greenland Ice Sheet, we analysed the morphology and mineral composition of dust in an ice core from northeastern Greenland (East Greenland Ice-Core Project, EGRIP), representing the period from 1910 to 2013, using scanning electron microscopy and energy-dispersive X-ray spectroscopy, and compared the results with those previously obtained for an ice core from northwestern Greenland (SIGMA-D). The composition of the SIGMA-D ice-core dust, comprising mostly silicate minerals, varied on a multi-decadal timescale due to an increased contribution of minerals originating from local ice-free areas during recent warming periods. In contrast, for the EGRIP ice-core dust, also consisting mostly of silicate minerals, there was relatively low compositional variation among the samples, suggesting that the mineral sources have not changed dramatically over the past 100 years. The subtle variation in the EGRIP ice-core mineral composition is likely due to a minor contribution of local dust. The type of silicate minerals differed significantly between the two ice cores; micas and chlorite, which form in cold dry regions, were abundant in the EGRIP ice core, whereas kaolinite, which forms in warm humid regions, was abundant in the SIGMA-D ice core. This indicates that the EGRIP ice-core dust likely originated from different geological sources than those for the SIGMA-D dust. A back-trajectory analysis indicated that the ice-core dust was transported from Northern Eurasia and North America to the EGRIP site, and that the contribution from each source was likely smaller and larger, respectively, than those for the SIGMA-D ice core. Furthermore, the higher illite content in the EGRIP ice core suggests dust transportation from Asian deserts. Although the back-trajectory analysis suggests that most of the air mass that arrived at the EGRIP site came from the Greenland coast, the mineral grain size and composition results showed that the local dust contribution was likely small.

## 1. Introduction

The history of dust deposition on the Greenland Ice Sheet has been reconstructed from ice cores, and provides substantial insight into past global climate and environmental changes. On geological timescales (e.g., from the Eemian to the Holocene), ice-core dust records show a strong correlation with climate variability, as indicated by $\delta^{18}O$ records. Dust concentrations in central Greenland ice cores increased by a factor of 100 during the Last Glacial Maximum (LGM) compared to those in the Holocene, and showed a strong correlation with temperature (Steffensen, 1997; Fuhrer et al., 1999;





Schüpbach et al., 2018). In the 20th century, the ice-core dust showed a seasonality characterized by a peak concentration in springtime (e.g., Drab et al., 2002). For the North Greenland Ice Core Project (NGRIP) ice core, the present-day dust concentration varies from >140 mg kg$^{-1}$ in the spring to >20 mg kg$^{-1}$ in the autumn (Bory et al., 2003a). This variability may

be related to the effects of climate change on dust sources and atmospheric transport (Svensson et al., 2000).

The Greenland ice-core dust content can also be used to predict darkening of the ice sheet. Recently, areas of "dark ice" have appeared and expanded on the ice-sheet surface (Shimada et al., 2016). Such ice is thought to be responsible for the recent reduction in the albedo of the Greenland Ice Sheet, and to have contributed to recent increase in the melting rate (e.g., Wientjes et al., 2011; Alexander et al., 2014; Shimada et al., 2016). One possible reason for the expansion of dark-ice

regions is an increase in the content of cryoconite, a mixture and/or aggregate of mineral dust and organic matter produced by glacial microbes (e.g., Takeuchi et al., 2014; Chandler et al., 2015). Mineral dust deposited on the glacial surface is likely to be related to increased cryoconite content because such dust is the principal constituent of cryoconite and can affect microbial biomass, another component of cryoconite, by supplying nutrients. Nagatsuka et al. (2016) reported that mineral dust deposited in the accumulation zone travelled through the ice sheet and outcropped again in the ablation zone. Such dust

is important for evaluating the formation of cryoconite and darkening of the Greenland Ice Sheet surface. This dust was probably widely deposited on the ice sheet in the past and then preserved in ice cores. Thus, determining the composition and sources of ice-core minerals exposed on the ice sheet surface is essential for evaluating the impact of dust on future darkening events.

In Greenland ice-core studies, possible dust sources have primarily been identified using geochemical analyses, particularly

those based on Sr, Nd, and Pb isotope ratios. The isotope ratios for mineral dust in central Greenland ice cores obtained by the Greenland Ice Sheet Project Two (GISP2) and the Greenland Ice Core Project (GRIP) indicate that the most likely sources for this dust are East Asian deserts and/or central European loess (Biscaye et al., 1997; Svensson et al., 2000; Újvári et al., 2015), and that North Africa is another potential source (Újvári et al., 2022). On the other hand, the isotope ratios for minerals contained in an ice core obtained in southeastern coastal Greenland suggest that the dust originated from local

sources during the Holocene and the last interglacial period (Eemian) (Simonsen et al., 2019). However, isotope analyses require large sample volumes, and thus mainly target ice-core dust from glacial periods with a high dust concentration. Several studies have attempted to measure isotopic ratios for ice-core dust from the Holocene, which is the present interglacial period characterized by a low dust concentration; however, a single sample required tens to thousands of years of ice accumulation (e.g., Bory et al., 2003a; Han et al., 2018; Simonsen et al., 2019). Therefore, little is known about sources

of mineral dust in interglacial periods, in which the dust concentration is low. To tackle this problem, Nagatsuka et al. (2021) analysed the morphology and mineralogy of individual dust particles in a northwestern Greenland ice core using scanning electron microscopy (SEM) and energy-dispersive X-ray spectroscopy (EDS), and produced historical records of mineral dust sources for the past 100 years with just 5-year resolution, thus demonstrating the effectiveness of this approach for determining variations in the sources of ice-core mineral dust during recent periods of low dust concentration.

Ice-core dust sources probably vary geographically even within the Greenland Ice Sheet because environmental conditions, such as altitude and distance from the coast, differ greatly among regions. The interior of the ice sheet generally has the highest elevation (Bamber et al., 2013; Helm et al., 2014). Bory et al. (2003b) reported that dust transported over long distances during the 11th and 17th-18th centuries was deposited at almost all elevated interior sites on the ice sheet, whereas it was likely that only dust from local ground sources was present at coastal sites. This distinction seems to depend on

distance from the ice sheet margin and/or altitude (Bory et al., 2003b). Furthermore, atmospheric transport may also affect the geographical variations in mineral dust sources over the ice sheet. Based on the compositional variations of ice-core minerals and a back-trajectory analysis, the air mass at the northwestern ice sheet (SIGMA-D site) originated from the west coast of Greenland and northern Canada during the period 1958–2014 (Nagatsuka et al., 2021). On the other hand, the air mass and dust at the central ice sheet (GISP2, GRIP, and NGRIP sites) likely originated mostly from Asia, Europe, and





North Africa during the period 45–12 kyr BP as well as the 17th century (Svensson et al., 2000; Bory et al., 2003b; Újvári et al., 2015, 2022). The primary source of dust in southeastern Greenland (Dye-3 site) was estimated to be Asia, with North Africa as an additional source in the 18th century (Lupker et al., 2010). Although differences in geographical conditions may have led to nonuniform dust deposition over the ice sheet, spatial variations in the continuous records for the Greenland ice-core dust sources have still not been clarified.

This paper describes temporal variations in the sources of mineral dust contained in an ice core obtained from the interior of the northeastern Greenland Ice Sheet covering almost 100 years (1910–2013). The East Greenland Ice-Core Project (EGRIP), which is an international ice coring project spearheaded by the University of Copenhagen, Denmark, was launched in 2015 as the first deep ice coring project in the northeastern inland ice sheet (Goto-Azuma et al., 2021). In the present study, we analysed the size and mineral composition of dust particles in a shallow ice core obtained at the EGRIP site using SEM and EDS. To identify regional variations in ice-core dust sources within the ice sheet, we compared the results to those for the northwestern coastal Greenland ice core (SIGMA-D) analysed by Nagatsuka et al. (2021). Based on the results obtained in the present study, temporal and geographic variations in ice-core dust sources are discussed.

## 2. Samples and analytical methods

### 2.1 EGRIP ice core

The ice core was drilled at an elevation of 2708 m above surface level on the northeast side of the EGRIP deep drilling site (75.38° N, 36.00° W) in July 2017 (Fig. 1). The EGRIP site is located 470 km west of the eastern coast of Greenland and close to the onset of the Northeast Greenland Ice Stream, which is the largest ice stream on the Greenland ice sheet (Joughin et al., 2010; Vallelonga et al., 2014). The ice core was recovered from a depth of 1.51–133.09 m.

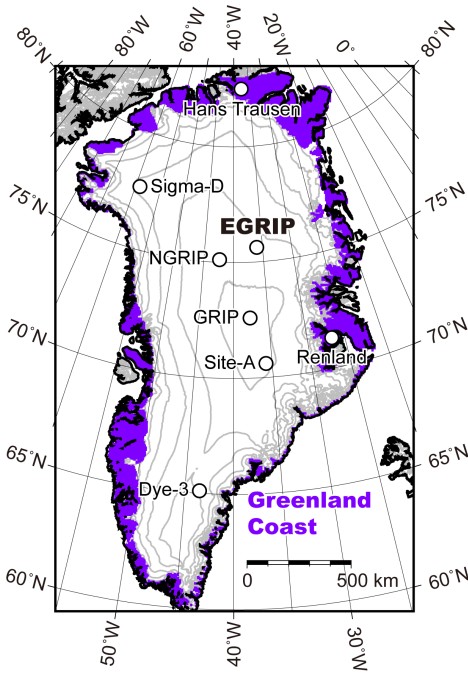

**Figure 1.** Location map of EGRIP ice core site (75.38° N, 36.00° W, 2708 m above surface level) and SIGMA-D, Hans Tausen, NGRIP, GRIP, Renland, Site-A, and Dye-3 ice core sites used for back-trajectory comparison, respectively. Contour lines are drawn at 500-m intervals. The purple shaded region denotes the ice-free coastal terrain (GrC).



### 2.2 Analyses of sodium, sulfate, tritium, and dust particles

To date the EGRIP ice core, the sodium (Na), sulfate ($SO_4$), and tritium concentrations were measured below a depth of 1.51 m. The Na and S concentrations were determined continuously along the core using an inductively-coupled plasma mass spectrometry (ICP-MS; 7700, Agilent Technologies, USA) system connected to a continuous-flow analysis (CFA) system at the National Institute of Polar Research (NIPR). $SO_4$ concentrations were calculated from S concentrations under the assumption that the dominant S-containing species is $SO_4$.

For tritium measurements, ice-core samples were taken at 0.04 m intervals at depths of 13–15 m. The tritium content was measured using a liquid scintillation counting technique at Nagoya University. The ice samples were distilled and mixed with a liquid scintillation cocktail (Ultima Gold LLT, PerkinElmer, USA). Radioactivity associated with tritium was measured using a low-background liquid scintillation counter (Quantulus 1220, PerkinElmer, USA) for 1500 min. The lower detection limit was about 7 TU, and the relative uncertainty was less than 5%.

For dust-particle size and concentration measurements, a portion of the meltwater was collected from the CFA samples taken at depth intervals of 0.12 m using a fraction collector. The concentration of dust was measured using a Coulter counter (Multisizer 4e, Beckman Coulter, USA) at the NIPR. The size bins covered the range of 0.7 to 18.0 µm.

### 2.3 SEM-EDS analysis of mineral dust

To extract mineral dust from the ice core, meltwater samples collected from the CFA samples using a fraction collector were placed in pre-cleaned glass vials at 10-year intervals. 500 µL of each sample was filtered through a polycarbonate membrane filter (0.2-µm pore size; 25-mm diameter; Advantec) in a Class 10000 clean room. We determined the size and chemical composition of individual mineral dust particles using SEM (Quanta FEG 450, FEI) and EDS (X-Max 50, Oxford Instruments, UK) at the NIPR. The filter was attached to aluminium stubs using carbon tape and coated with a layer of platinum. A total of 200 random particles from the filter was observed and the equivalent circle diameter was measured using image-processing software (ImageJ, National Institutes of Health, USA). The contents of major elements and related

oxides were determined from the EDS spectra. Details of the SEM-EDS analytical method are described in Nagatsuka et al. (2021).

### 2.4 Mineral identification

Mineralogical identification of the EGRIP ice-core dust was performed using three procedures (Nagatsuka et al., 2021): (1)

matching of the spectral pattern for each particle to those for mineral standards (Severin, 2004); (2) comparison of the morphology and oxide composition of the ice-core dust to those of mineral standards; (3) application of the elemental peak intensity ratio sorting scheme used to identify Greenland ice-core (GISP2) mineral dust (Donarummo et al., 2003). Based on the results of previous studies, a comparison of the results obtained using these procedures enables reliable identification of ice-core minerals (Maggi et al., 1997; Wu et al., 2016).

Most of the silicate minerals in the EGRIP ice core were categorized into five types (Types A–E) based on the formation environment, formation process, and possible sources, as performed for the SIGMA-D ice core. Details of each mineral type are given in Table 1. These minerals have localized distributions (Ito and Wagai, 2017). The relative abundance for pairs of mineral types reflects the relative contributions of chemical and physical weathering processes, and is thus an indicator of latitude (e.g., Biscaye, 1965; Griffin et al., 1968; Biscaye et al., 1997; Maggi, 1997; Svensson et al., 2000; Donarummo et

al., 2003). The variations in mineral composition among the five mineral types reflect the climatic and geological conditions in their source areas, and can be therefore used as indicators for the dust source and the transportation process in different periods.





**Table 1.** Description of silicate mineral types in the EGRIP ice core


| Type | Minerals | Notes | References | Possible sources |
|---|---|---|---|---|
| Type A | Kaolinite and other kaolin minerals (nacrite, dickite, halloysite) and/or pyrophyllite | Clay minerals composed of Si and Al generally formed in warm and humid regions by chemical weathering. | Mueller and Bocquier, 1986; Velde, 1995; Bergaya et al., 2006; Nagatsuka et al., 2021 | Low- to middle-latitude areas (e.g., Central Africa, South America, Southeast Asia) and relict deposits of past warmer climates (e.g. Tertiary Northern Canadian deposits) |
| Type B | Micas, chlorite, and their mixture | Clay minerals formed in cold and dry regions by mechanical weathering. | Cremaschi, 1987; Pye, 1987; Velde, 1995 | High-latitude (e.g., North America, Russia, North Europe, Greenland) and/or desert areas (e.g., Asia and North Africa) |
| Type C | Feldspars (Na/Ca/K-plagioclase and K-feldspar) | Minerals formed in cold and dry regions by mechanical weathering. | Nahon, 1991; Bory et al 2003b; Nagatsuka et al., 2014 | High-latitude (e.g., North America, Russia, North Europe, Greenland) and/or desert areas (e.g., Asia and North Africa) |
| Type D | Mafic minerals | Minerals formed by mechanical weathering and less common in atmospheric dust. | Deer et al., 1993 | Local areas (Greenland) |
| Type E | Quartz | Most resistant to weathering processes at the Earth's surface and whose abundance in the atmosphere is likely related to desert source areas. | Pye, 1987; Yokoo et al., 1994; Genthon and Armengaud, 1995) | Desert areas (e.g., Asia and North Africa). |



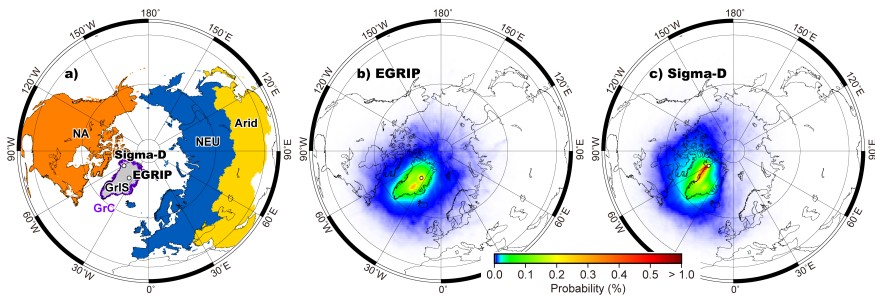

**Figure 2.** Map showing (a) location of EGRIP and SIGMA-D ice core sites in Greenland and five regions used for calculating regional contribution (GrIS: Greenland Ice Sheet, grey; GrC: Greenland coast, purple; NA: North America, orange; NEU: Northern Eurasia, blue; Arid Asia: mid-latitude arid regions consisting of East and Central Asia, and Middle East, yellow) and probability distribution for air mass at (b) EGRIP and (c) SIGMA-D sites from 7-day three-dimensional back trajectory analysis from 1958 to 2014.

## 2.5 Back-trajectory analysis

To investigate the potential dust sources for the EGRIP site, air mass transport pathways were analysed by using the Hybrid Single-Particle Lagrangian Integrated Trajectory (HYSPLIT) model (Stein et al., 2015). The initial air masses were set at 50, 500, 1000, and 1500 m above ground level at the EGRIP site for the 20-day back trajectories. The probability distribution for the air mass within this altitude range was calculated with $1° \times 1°$ degree resolution. We assumed that the dust particles had been deposited via a wet process (Iizuka et al., 2018; Parvin et al., 2019), for which the daily precipitation at the time when

the air mass arrived at the ice-core site was considered (Nagatsuka et al., 2021). We used daily precipitation from the ERA5 reanalysis dataset produced by the European Centre for Medium-Range Weather Forecasts (Hersbach et al., 2020). The regional contribution was also calculated from the probability distribution, for which land areas were divided into five regions; the Greenland Ice Sheet (GrIS), the Greenland coast (GrC), North America (NA), Northern Eurasia (NEU), and mid-latitude arid regions (Arid: consisting of East and Central Asia and the Middle East) (Fig. 2a). Considering the dust

source, the ocean and GrIS were excluded from the calculation.

## 2.6 Snow cover fraction

To examine the surface conditions for adjacent source areas of mineral dust in the EGRIP ice core, we analysed the changes in inter-annual snow cover fraction obtained from numerical simulations using a climate model. Global climate models have been used by international organizations when performing numerical simulations to reproduce or predict past or future

climate change. The results obtained in these studies have been compiled and published by World Climate Research Programme Coupled Model Intercomparison Project Phase 6 (CMIP6; Eyring et al., 2016). To investigate sources for the EGRIP dust, we used snow cover fractions derived from four land historical simulations with four different reanalysis datasets as atmospheric forcing for the period 1850–2014. The reanalysis datasets, namely, Princeton (Sheffield et al., 2006), WFDEI (Weedon et al., 2014), GSWP3 (Kim, 2017), and CRUJRA (Harris, 2019), provided the meteorological conditions

for the Land Surface, Snow and Soil moisture Model Intercomparison Project (LS3MIP; van den Hurk et al., 2016) of CMIP6. In this study, we analysed the snow cover fractions simulated with a climate model MIROC6 (Tatebe et al., 2019) in LS3MIP historical experiments (Onuma and Kim, 2020a, b, c, d). Data for inter-annual variations in snow cover fraction during summer on the northeast and southeast coasts of Greenland (boundary at 70° N) were obtained in the same manner as in Nagatsuka et al. (2021).



## 3. Results

### 3.1 Dating of EGRIP ice core

Ice-core dating was performed by counting annual layers of Na concentration, which showed clear seasonal variations (Fig. A1). The observed seasonality of the water stable isotope ratio and chemical components in the ice cores and snowpack has been reported previously at various sites on the Greenland Ice Sheet and is often used for annual layer counting (Whitlow et al., 1992; Legrand and Mayewski, 1997; Kuramoto et al., 2011; Oyabu et al., 2016; Kurosaki et al., 2020; Komuro et al., 2021; Nakazawa et al., 2021; Nagatsuka et al., 2021; Sinnl et al., 2022). The winter season in the EGRIP ice core was defined as the depth at which the Na concentration had a maximum value, and we counted winter season to winter season as 1 year.

Three fixed dates were provided by the ice layers, tritium profile and $SO_4$ spikes. The ice layer at a depth of 2.14 m is assumed to correspond to the 2012 summer based on a snow pit observation at EGRIP in summer 2017 (Komuro et al., 2021). We interpreted the ice layer at a depth of 2.40 m as a layer that formed by refreezing of meltwater produced by surface melting in summer 2012. The sharp tritium peak at a depth of 14.1 m corresponds to the nuclear-bomb testing in 1963 (Koide et al., 1982; Clausen and Hammer, 1988). The two sharp $SO_4$ peaks at depths of 23.18 and 23.30 m are presumed to be associated with the eruption of the Katmai volcano, Alaska, which occurred on June 6–8, 1912 (e.g., Hildreth, 1983; Hildreth and Fierstein, 2000;Sinnl et al., 2022). We assigned the latter peak to this eruption because this peak appears after the Na concentration peak (winter) and corresponds to the timing of the eruption in early summer. The $SO_4$ peak at a depth of 23.18 m, which is assumed to correspond to the 1913 winter/spring, is interpreted as stratospheric fallout of sulphate aerosols produced by the same eruption, although it may also be derived from anthropogenic sulphate, peaking in winter to early spring (Beer et al., 1991, Kuramoto et al., 2011; Oyabu et al., 2016) and/or the eruption of the Hekla volcano, Iceland, starting on 24 April 1913 (Bigler et al., 2002). Volcanic aerosols from the 1912 Katmai eruption, known as the world's largest 20th-century volcanic eruption, were injected into the stratosphere and remained suspended until at least as late as December 1914 (e.g. Volts, 1975a and 1975b; Hildreth, W. and Fierstein, J., 2012; Burke et al., 2019). Volcanic sulphate deposits from the stratosphere are also found in the 1913 annual layer of other Arctic ice cores, and are identified by a subsidiary $SO_4^{2-}$ peak (Yalcin et al., 2007; Burke et al., 2019). From a comparison of the annual layer counts and these reference horizons, we estimated that the ice-core dating uncertainty is 1 year. We also compared our EGRIP chronology to the Greenland ice-core chronology 2021 (GICC21, Sinnl et al., 2022) based on multiple ice cores. Both chronologies are in good agreement within their respective uncertainties.

From these analyses, we estimated that a depth of 23.64 m corresponds to the year 1910 and that the average accumulation rate for the period 1910–2013 was 0.11 m w.e. yr$^{-1}$. This is consistent with that for the period 1607–2011 at EGRIP reported by Vallelonga et al. (2014) (also 0.11 m w.e. yr$^{-1}$). Based on our chronology, the dates associated with the $SO_4$ peaks at 2.56, 9.20, and 23.17 m were estimated to be 2011, 1987, and 1913, respectively. The existence of volcanic signals in these years is consistent with the results of previous ice-core studies at EGRIP (Kjær et al., 2016; Kjær et al., 2022).



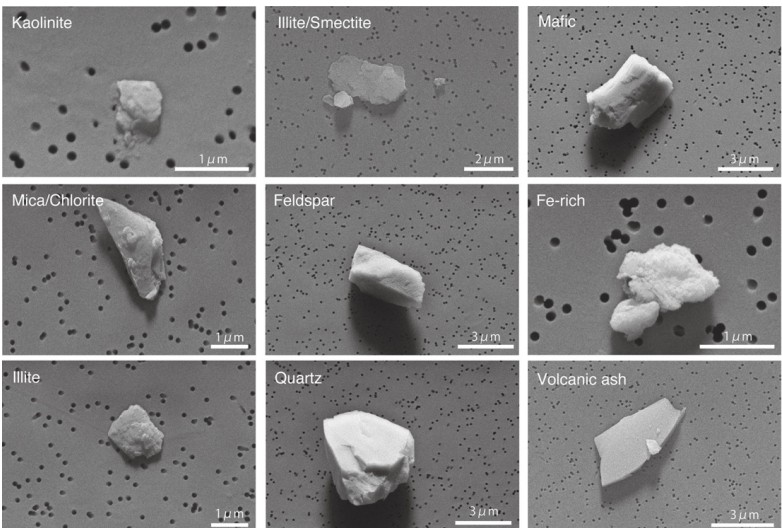

**Figure 3.** SEM micrographs of samples from each mineral group in EGRIP ice core.

## 3.2 Particle morphology

Figure 3 shows SEM images of various types of dust particles observed in the EGRIP ice core. The number size distribution
indicates that the diameter of most particles was smaller than 2 μm (Table 2, Fig. 4), similar to those from other Greenland
ice cores (e.g., Steffensen, 1997; Biscaye et al., 1997; Nagatsuka et al., 2021). The mean particle diameter, calculated as 10-
year averaged values, was 0.85–1.36 μm, with a single modal distribution with a peak at 0.37–0.57 μm. The maximum
particle diameter was 5.06–8.94 μm. The size distribution varied depending on the period, showing a narrow peak with a fine
mode (0.37–0.43 μm) for the 1920–1960 samples and a slightly broad peak with a coarser mode for the 1970–2013 samples
(0.42–0.48 μm; Fig. A2). The mode values showed an increasing trend over the 100-year period except for the 1910–1920
sample that had the largest number of coarse particles. No samples were found to contain particles larger than 10 μm.

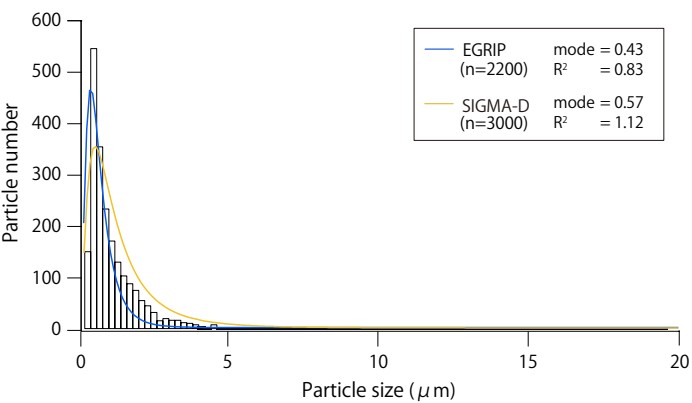

**Figure 4.** Size distribution and log-normal fitting results (mode: mode diameter; $R^2$: half peak width) for minerals in EGRIP ice core and
SIGMA-D ice core (fitting results only) samples during periods of 1910 to 2013 and 1915 to 2013, respectively.



**Table 2.** Description of EGRIP ice core dust samples

| Period | Ice | | Dust (particle size) | | |
|---|---|---|---|---|---|
| | Top (m) | Bottom (m) | Average (µm) | Maximum (µm) | Log-normal mode (µm) |
| 1910−1920 | 21.95 | 23.64 | 1.36 | 8.25 | 0.57 |
| 1920−1930 | 20.28 | 21.95 | 0.96 | 8.55 | 0.43 |
| 1930−1940 | 18.25 | 20.28 | 0.89 | 5.59 | 0.41 |
| 1940−1950 | 16.55 | 18.25 | 0.95 | 8.27 | 0.37 |
| 1950−1960 | 14.80 | 16.55 | 0.88 | 7.60 | 0.37 |
| 1960−1970 | 12.60 | 14.80 | 1.10 | 8.29 | 0.48 |
| 1970−1980 | 10.79 | 12.60 | 0.85 | 7.78 | 0.42 |
| 1980−1990 | 8.54 | 10.79 | 1.06 | 5.06 | 0.44 |
| 1990−2000 | 5.84 | 8.54 | 1.06 | 6.94 | 0.46 |
| 2000−2010 | 3.01 | 5.84 | 1.07 | 8.94 | 0.45 |
| 2010−2013 | 1.93 | 3.01 | 0.99 | 8.72 | 0.48 |

### 3.3 Quantitative estimation of mineral dust

The EDS analysis showed that for all samples the ice-core dust was composed predominantly of silicate minerals (94–98%, Fig. 5). The silicates were categorized as quartz, mafic minerals, Na/Ca-, K-, and Na/K-feldspars, clays (kaolinite, other kaolin minerals (nacrite, dickite, halloysite) and/or pyrophyllite (hereafter referred to as high Si + Al minerals), smectite, illite, micas, and chlorite, as well as mixed layers of illite/smectite and micas/chlorite) (Figs. 3 and 6). We also observed a few minerals with an identical rhyolitic volcanic glass composition, which have also been found in the other Greenland ice
cores (e.g., Cook et al., 2022).

Based on the semi-quantitative EDS analysis, the proportion of the mica/chlorite mix was the highest (21–37%) and that of smectite was the lowest (0–2%) for all silicate mineral particles in nearly every period (Table 3 and Fig. 6). The proportions of high Si + Al minerals, illite/smectite mix, illite, and quartz were the second highest, varying in the ranges 9–22%, 4–14%, 3–16%, and 4–12%, respectively.

The silicate mineral composition varied among the samples. Lower high Si + Al minerals content and higher illite/smectite mix and illite content were found for the 1910–1970 samples (high Si + Al minerals: 9–14%, illite/smectite mix: 6–14%, illite: 9–16%) compared to the 1970–2013 samples (high Si + Al minerals: 18–22%, illite/smectite mix: 4–5%, illite: 3–13%). A higher volcanic glass content (11%) was also found in the 1910–1920 sample than in samples from other periods (0–2%). The compositions of other silicate minerals were similar for the same periods.

The ice-core samples also composed of non-silicate minerals at low concentrations. They consisted mainly of an Fe-dominant mineral, identified as an Fe oxide (hematite, magnetite, or pyrite, Fig. 5). The relative abundance of these minerals was 2–6%, and showed low variation among the samples.





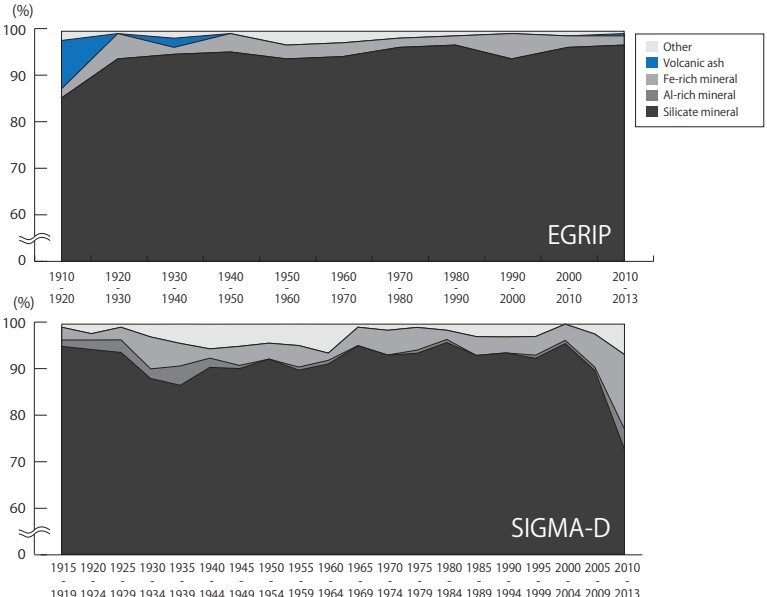

**Figure 5.** Variations of insoluble mineral records in EGRIP ice core (top) and SIGMA-D ice core (bottom) dust with 10- and 5-year resolutions, respectively.

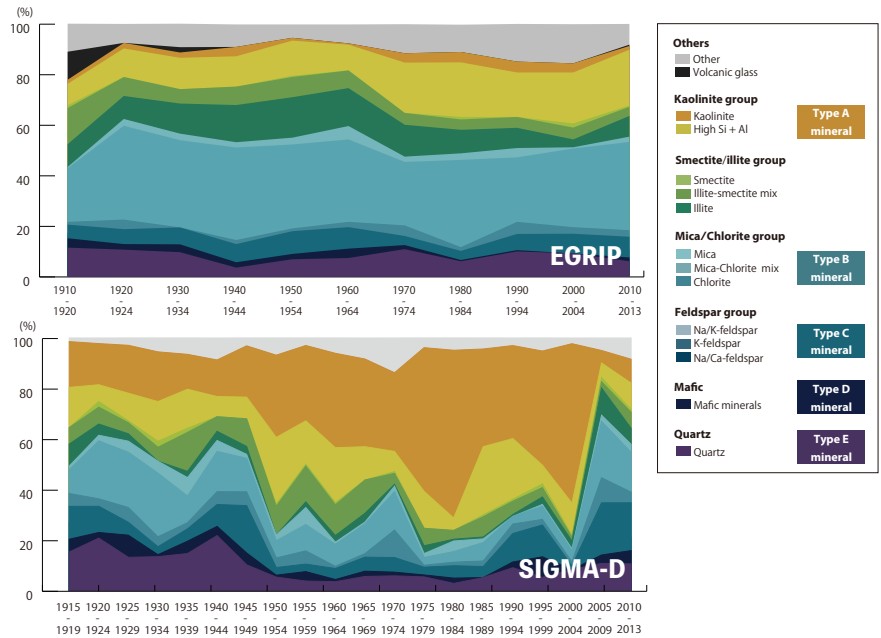

**Figure 6.** Variations of silicate mineral compositions in EGRIP ice core (top) and SIGMA-D ice core (bottom) dust with 10- and 5-year resolutions, respectively. "Feldspars" includes Na/Ca-feldspar and K-feldspar.





**Table 3.** Relative abundance (%) of silicate mineral groups for each EGRIP sample

| Sample period | Kaolinite | High Si+Al | Smectite | Illite/Smectite | Illite | Micas | Micas/Chlorite | Chlorite | Na/Ca-feldspar | K-feldspar | Na/K-feldspar | Mafic | Quartz | Volcanic glass | Unknown |
|---|---|---|---|---|---|---|---|---|---|---|---|---|---|---|---|
| 1910–1920 | 1.6 | 8.5 | 1.1 | 14.3 | 9.0 | 0.5 | 21.2 | 1.1 | 3.2 | 1.1 | 1.1 | 3.7 | 11.6 | 11.1 | 11.1 |
| 1920–1930 | 2.2 | 11.3 | 0.0 | 7.5 | 9.1 | 2.7 | 37.1 | 3.8 | 4.3 | 1.6 | 0.0 | 2.2 | 10.8 | 0.0 | 7.5 |
| 1930–1940 | 2.1 | 12.6 | 0.0 | 5.7 | 11.9 | 2.6 | 34.5 | 0.0 | 5.2 | 1.0 | 0.5 | 3.1 | 9.8 | 2.1 | 9.3 |
| 1940–1950 | 3.7 | 12.0 | 0.0 | 7.3 | 14.7 | 2.1 | 36.6 | 1.6 | 3.1 | 3.1 | 1.0 | 2.1 | 3.7 | 0.0 | 8.9 |
| 1950–1960 | 1.1 | 13.9 | 0.5 | 8.0 | 16.0 | 2.7 | 33.2 | 1.1 | 8.0 | 0.5 | 0.5 | 2.1 | 7.0 | 0.0 | 5.3 |
| 1960–1970 | 0.5 | 10.2 | 0.0 | 7.0 | 15.0 | 5.3 | 32.6 | 2.1 | 4.3 | 2.1 | 2.1 | 3.7 | 7.5 | 0.0 | 7.5 |
| 1970–1980 | 3.7 | 19.9 | 0.0 | 4.7 | 12.6 | 2.1 | 25.1 | 4.2 | 2.6 | 0.0 | 1.0 | 1.6 | 11.0 | 0.0 | 11.5 |
| 1980–1990 | 4.1 | 21.6 | 1.0 | 4.1 | 9.3 | 2.6 | 34.5 | 1.5 | 2.6 | 1.0 | 0.0 | 0.5 | 6.2 | 0.0 | 10.8 |
| 1990–2000 | 4.3 | 17.6 | 0.0 | 4.3 | 8.0 | 3.7 | 25.5 | 4.8 | 4.8 | 1.6 | 0.0 | 0.5 | 10.1 | 0.0 | 14.9 |
| 2000–2010 | 3.6 | 20.2 | 1.6 | 4.7 | 3.1 | 0.5 | 31.1 | 2.6 | 6.2 | 0.0 | 1.6 | 0.0 | 9.3 | 0.0 | 15.5 |
| 2010–2013 | 1.5 | 22.1 | 0.5 | 3.6 | 8.2 | 2.1 | 34.9 | 2.6 | 5.1 | 2.6 | 0.5 | 1.5 | 6.2 | 0.5 | 8.2 |





### 3.4 Source regions for EGRIP ice-core dust

Figure 2b shows the averaged probability distributions for an air mass arriving at the EGRIP site from 1958 to 2014, calculated from the 7-day back-trajectories. Figures 7 and 8 show the regional contributions to air mass arriving at EGRIP (Fig. 7b and blue lines in Fig. 8) and Sigma-D (Fig. 7b and yellow lines in Fig. 8), in terms of inter-annual variability (Fig. 7) and backward temporal change (Fig. 8), respectively. The HYSPLIT back-trajectory model suggests that the air mass mainly originated from Greenland (Figs. 2b, 7a and 8a). Excluding the ice sheet and ocean areas, which are not possible mineral dust sources, the air mass have originated mainly from the Greenland coast (56–67%) as well as from North America (20–31%) and Northern Eurasia (6–20%, Figs. 8b and 8c). There is seen to be little inter-annual variation in the contributions from these three source regions.

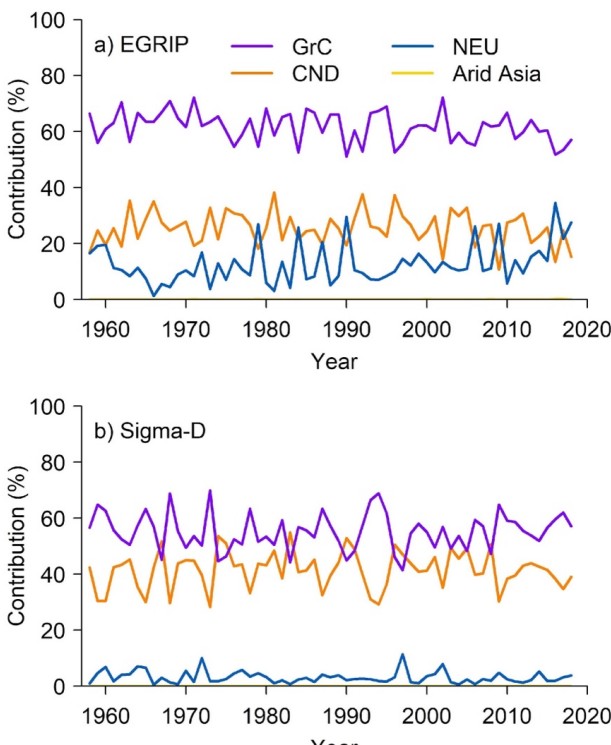

**Figure 7.** Annual variations in regional contribution of air mass to (a) EGRIP and (b) SIGMA-D sites excluding ice sheet and ocean areas. GrC, CND, NEU, and Arid Asia denote the ice-free Greenland coastal region (Fig. 1), Canada, Northern Eurasia, and mid-latitude arid regions, respectively (Fig. 2a).





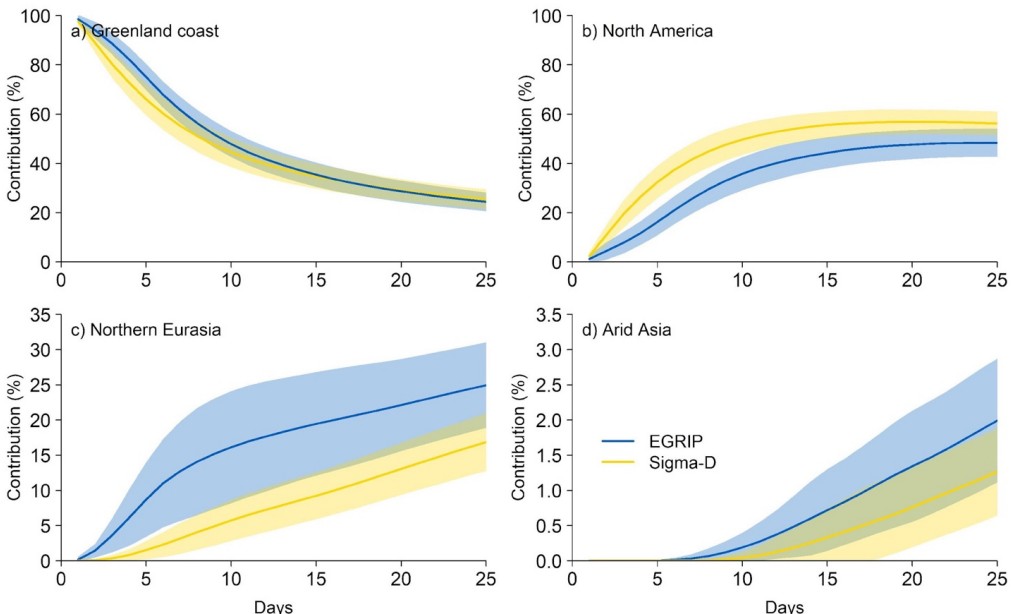

**Figure 8.** Air mass contribution for 25-day back trajectory from (a) Greenland coast (GrC), (b) Northern Eurasia (EU and Russia), (c) North America (Canada and US), and (d) Arid Asia to EGRIP and SIGMA-D sites.

## 4. Discussion

### 4.1 Variability of mineral composition

The EGRIP ice-core dust samples from 1910 to 2013 mainly consisted of silicate minerals, the most common mineral groups
in the Earth's crust (Deer et al., 1993), with small amounts of Fe-dominant minerals, as was the case for the SIGMA-D ice core (Fig. 5). We did not identify any carbonate minerals in the EGRIP core, which is probably due to the sample preparation process. The EGRIP ice core was melted and filtered, whereas the SIGMA-D ice core was freeze-dried. Thus, carbonate minerals in the EGRIP ice core were probably dissolved in the melted ice samples. Comparing the proportion of ice-core minerals other than carbonate minerals, there were no significant differences in the silicate and Fe-dominant minerals
between the two ice cores (EGRIP: 86–97% and 2–6%, SIGMA-D: 73–96% and 1–16%, respectively). The lack of Al-dominant minerals in the EGRIP ice core might be due to differences in the geological sources.

The variation in the relative abundance of different silicate minerals in the EGRIP ice core differed among mineral types (Figs. 6 and 9). Since 1970, the relative abundance of illite has decreased but that of high Si + Al minerals (Type A) has increased, implying multiple geological sources with varying relative contributions.




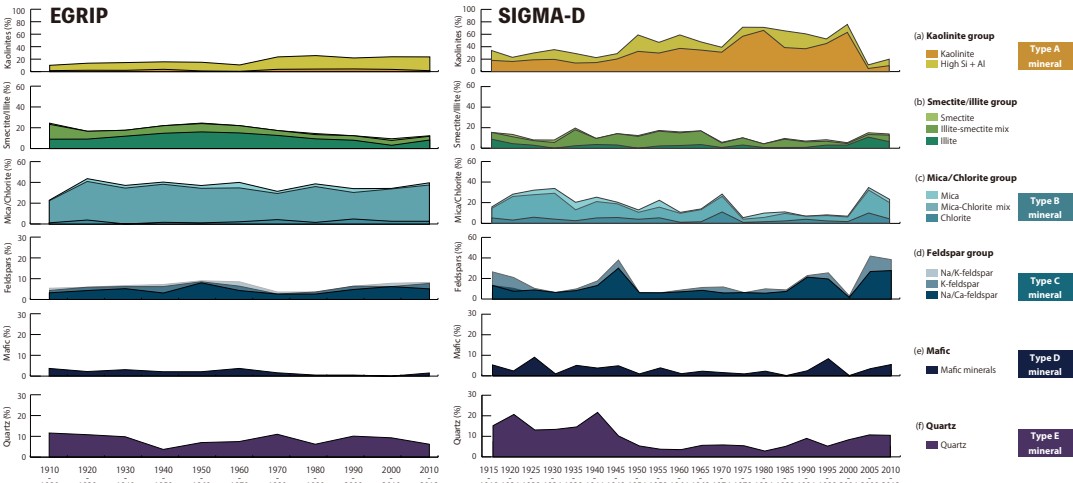

**Figure 9.** Comparison of historical records in silicate mineral proportions from EGRIP and SIGMA-D ice cores (a-f, a: kaolinite group, b: smectite group, c: micas/chlorite group, d: feldspar group, e: mafic mineral, f: quartz).

The EGRIP and SIGMA-D ice cores exhibit significantly different variations in silicate composition. For the SIGMA-D core, multi- and inter-decadal compositional variations are observed, indicating that the ice-core dust originated from different geological sources during the past 100 years (Nagatsuka et al., 2021). In contrast, the EGRIP mineral composition has low variation among the samples (Figs. 6 and 9), suggesting that the sources did not change dramatically during this period. A possible cause of the subtle variations in the EGRIP ice-core mineral composition is a minor contribution from sediments on the Greenland coast (local). The multi-decadal variation of the SIGMA-D ice-core dust, which is strongly affected by Greenland temperature changes, was caused by an increasing contribution of minerals that originated from local ice-free areas (Type B, C, D, and E minerals) because snow/ice cover duration in the Greenland coastal region was shorter during the two warming periods of 1920s–1950s and 2000–2013 (Nagatsuka et al., 2021). The EGRIP ice core contained lower proportions of Type C, D, and E minerals (4–9%, 0–4%, and 4–12%, respectively) compared to those for the SIGMA-D ice core (2–21%, 0–9%, and 3–22%, respectively). It has been suggested that dust sources depend on the distance of ice-core sites from the ice-sheet margin and/or the altitude of the site, and that the majority of the dust deposited at interior sites (NGRIP, GRIP, Site-A, and Dye-3) is associated with long-range transport from distant deserts, whereas at coastal sites (Hans Tausen and Renland), the primary sources are local ice-free areas (Bory et al., 2003b) . The EGRIP site is located in the interior, whereas the SIGMA-D site is on the coast. Thus, the local dust contribution to the EGRIP ice core is smaller and the compositional variation likely reflects this.

The morphological properties of the EGRIP ice-core dust also suggest a small supply of minerals from local source areas. The particle size distribution for the EGRIP dust show lower mean, maximum, and modal diameters with little variation compared to those for the SIGMA-D ice core, except for the samples from 1970 to 1980 (Table 2, Fig. A2). Simonsen et al. (2019) suggested that particles with diameters smaller than 2 μm and larger than 8 μm in Greenland ice cores can be used as an indicator of distant and local sources of mineral dust, respectively. The SIGMA-D ice core contained coarse particles (larger than 8 μm), especially in the warming periods when the local dust supply increased (1–9 particles in each sample, a total of 150 particles, from 1915 to 1949 and from 2000 to 2013). On the other hand, there were few particles with a diameter of larger than 8 μm (a total of only 7 particles in all samples) in the EGRIP ice core; most of the particles were



smaller than 2 µm. These results imply that the EGRIP ice-core dust was mainly composed of particles transported over long distances.

**4.2 Possible sources of ice-core minerals**

The silicate mineral composition differed substantially between the two ice cores, indicating that the dust was likely transported from different geological sources. The EGRIP ice core showed significantly lower kaolinite (Type A) content

and higher micas/chlorite (Type B) and illite content (1–4%, 21–37%, and 3–16%, respectively, Figs. 6 and 9) compared to those for the SIGMA-D ice core (5–66%, 3–25%, and 0–11%, respectively). According to Nagatsuka et al. (2021), the SIGMA-D ice core was constantly supplied with kaolinite, mainly originating from northern Canada. The lower kaolinite content of the EGRIP ice core is consistent with that of other Greenland inland ice cores (GRIP, 4–16% (Svensson et al., 2000); GISP2, 0–2% (Donarummo et al., 2003)) and likely reflects the small contribution from northern Canada. On the

other hand, the dominance of the Type B minerals during the past 100 years indicates a constant supply of these minerals, likely originating from cold and dry high-latitude areas, to the EGRIP site.

The higher illite content in the EGRIP ice core than in the SIGMA-D ice core is probably due to the contribution of minerals from Asian deserts. Újvári et al. (2015) analysed the clay mineralogy of potential source areas for Greenland ice cores and showed that Alaskan, central European, Siberian, and especially Chinese loess has a higher illite proportion than the other

examined areas. There have also been reports of a large contribution of illite with a low contribution of smectite and a low kaolinite/chlorite ratio in Asian-sourced dust in the Greenland ice core and snow (e.g., Biscaye et al., 1997; Drab et al., 2002; Újvári et al., 2022). These mineralogical characteristics are consistent with the EGRIP samples. Furthermore, the decreasing trend for illite content in the EGRIP ice core since 1980 (Fig. 9) is similar to that for dust occurrence in Asian deserts. Dust events have decreased rapidly since the 1980s in East Asia, particularly in northern China, primarily because of changes in

meteorological conditions (Liu et al., 2020). The frequency of dust storms and dust weather in the 1950s–70s was about twice that after the mid-1980s (Qian et al., 2002). A ternary clay mineralogy diagram of the proportion of illite/micas/chlorite, kaolinite, and smectite shows that samples from EGRIP ice-core dust from 1910 to 1970 have a higher illite/micas/chlorite content than that for samples from 1980 to 2013, which is similar to Greenland ice core (GRIP and GISP2) dust and the loess and desert sand from Asia and North America, the potential sources of these ice-core minerals

(Fig. 10). These results indicate that minerals originating from Asian deserts seem to be the best candidate for the EGRIP illite source.

Previous studies have also reported silicate minerals other than clays in Greenland ice core (GRIP and GISP2) dust, mostly from glacial periods, and their possible sources (Biscaye et al., 1997; Maggi, 1997; Svensson et al., 2000; Újvári et al., 2022). The EGRIP ice core showed similar clay mineralogy to that of the GRIP and GISP2 ice cores, but showed slightly higher

Type A and significantly lower Type E (quartz) mineral content. Quartz possibly originated from desert areas, as described in the mineral identification method section (Table 2). In addition, distant deserts of Asia and Africa, are the primary sources of the minerals in the GRIP and GISP2 ice cores. Therefore, the lower quartz content may reflect smaller contributions from such desert areas to the EGRIP site compared to those for other ice-core sites. There is likely another main source for the EGRIP ice-core dust.




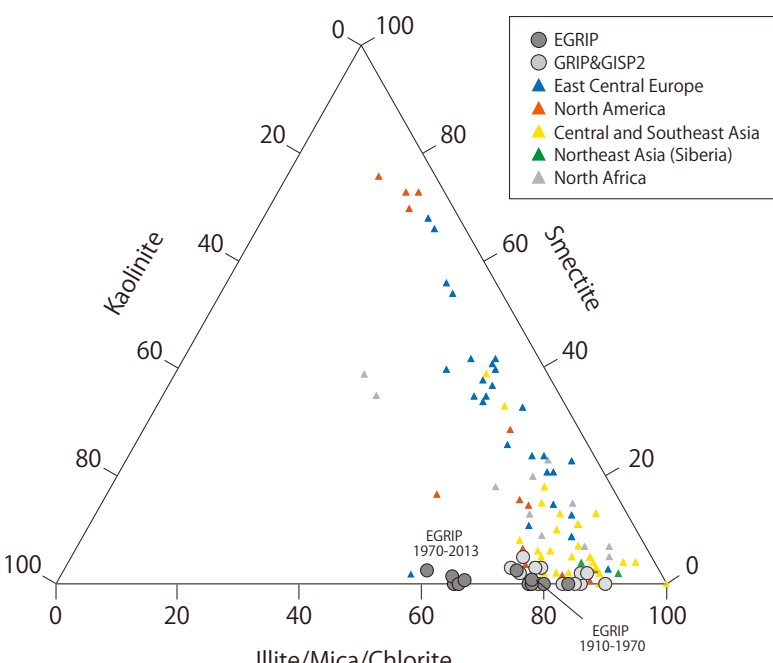

**Figure 10.** Ternary clay mineralogy diagram of EGRIP ice core dust during period of 1910 to 2013 and published literature data for Greenland Ice Sheet Project 2 (GISP2) Greenland Ice Core Project (GRIP) ice core dust during last glacial maximum (LGM) and potential source areas in illite/micas/chlorite-smectite-kaolinite space (GRIP and GISP2: Biscaye et al., 1997; Svensson et al., 2000, East Central Europe: Biscaye et al., 1997; Svensson et al., 2000; Újvári et al., 2012, 2015, 2022, Martinez-Lamas et al., 2020, North America: Biscaye et al., 1997; Svensson et al., 2000; Újvári et al., 2015, 2022, Central and Southeast Asia: Biscaye et al., 1997; Svensson et al., 2000; Újvári et al., 2015, 2022, Li et al., 2018, Northeast Asia: Újvári et al., 2015, 2022; North Africa: Elmouden et al.,2005; Skonieczny et al., 2011; Újvári et al., 2022).

The results in Figure 8 from the back-trajectory analysis reveal that most of the air mass at the EGRIP site between 1958 and 2014 originated from coastal Greenland and that a smaller proportion originated from North America and Northern Eurasia. These three regions have also been suggested as possible dust sources for the SIGMA-D ice core. However, the relative contribution from North America and Northern Eurasia significantly differed between the two ice-core sites. The contribution of air mass from North America to the EGRIP site was smaller than that to the SIGMA-D site, consistent with the lower proportion of Type A minerals in the EGRIP ice core. On the other hand, the larger contribution from Northern Eurasia indicates that this region seems to be one of the sources of Type B minerals (micas and chlorite). Although mineralogical results indicate that Asian deserts are also a possible source, our trajectory results show little contribution from these regions even in the 25-day back-trajectory. This is likely due to the inability of the back-trajectory analysis to identify dust transport from Asia to Greenland (Schüpbach et al., 2018). A slightly higher air mass contribution from Asia and arid regions to the EGRIP site suggests that the ice core contains more dust from these regions than does the SIGMA-D ice core (Fig. 8d). The trajectory results also indicate that the EGRIP ice-core dust was unlikely to have originated from African deserts despite the clay mineralogy, suggesting a possible contribution from this region. We can conclude that the contribution of air mass from local source areas (i.e., coastal Greenland) to both ice-core sites was large but that the mineral dust contribution from local source areas to the EGRIP site was relatively small.





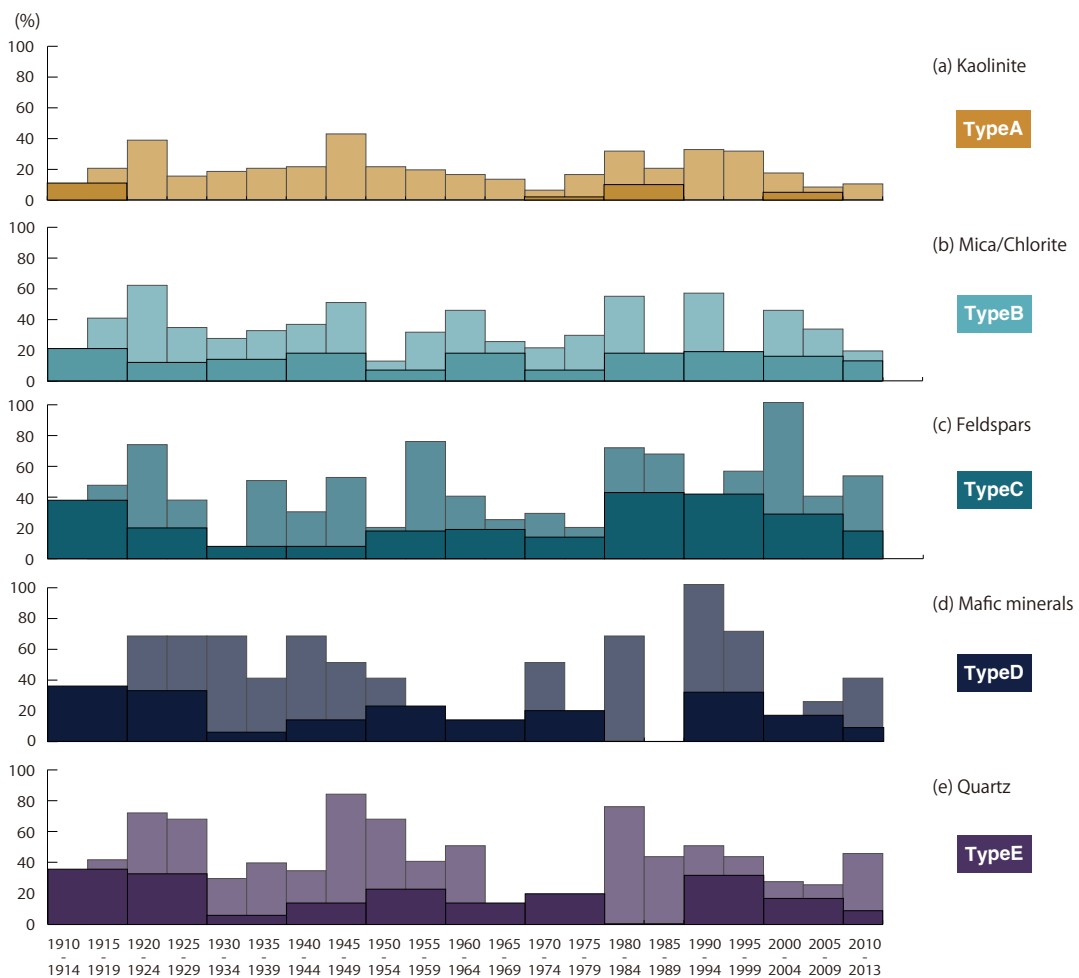

**Figure 11.** Historical changes in proportion of coarser fraction (larger than 2µm) of mineral particles in (a) kaolinite, (b) micas/chlorite, (c) feldspars, (d) mafic minerals, and (e) quartz. Dark color denotes EGRIP ice core dust and light color denotes SIGMA-D ice core dust.

Despite the smaller supply of locally sourced dust compared to that for the SIGMA-D ice core, as described in section 4.1, the SEM-EDS analysis indicated that the contribution of local dust particles in the EGRIP ice core may have increased slightly in recent years. The particle size depended on the ice-core dust mineralogy. The coarser ice-core dust fraction (larger than 2 µm) was rarely composed of clay minerals (Types A and B, Fig. 11), but instead consisted of a relatively large amount of Type C, D, and E minerals. This size dependency is consistent with dust in other Greenland ice cores, although the sample periods are different (Biscaye, 1965; Svensson et al., 2000; Nagatsuka et al., 2021). Type C minerals likely originated from high-latitude areas including the Greenland Coast and/or desert areas, and can be used as an indicator for the local dust contribution to the Greenland Ice Sheet because they have more abundant coarser particles than clay minerals (Bory et al., 2003a). The higher proportion of coarser particles in Type C minerals in the 1980–2013 samples (Fig. 9, 18–43%) than samples from other periods (8–20%, except for a high proportion in the 1910–1920 sample) is consistent with the warming

periods since the 1990s (Box et al., 2009) in Greenland, and thus likely reflects a recent increase in local dust transportation to the EGRIP site.





**Figure 12.** Historical changes in (a) annual average number concentration of EGRIP ice core dust with diameters of (a) 0.70–2.01 μm and (b) 2.01–6.49 μm measured using Coulter counter, (c) their ratios of coarse (2.01–6.49μm)/fine (0.70–2.01μm) particles, (d) the North Atlantic Oscillation index (NAO; Hurrell and National Center for Atmospheric Research Staff, 2021), and (e) snow cover fraction anomalies and (f) surface temperature anomalies in Greenland. The temperature records of Ittoqqortoormii and Tasiilaq in eastern Greenland located 750 km southeast and 1100 km south of the EGRIP site are from Cappelen (2019). Snow cover fraction anomalies deviate from the 1915–2012 average in NE and SE Greenland. Horizontal dashed red lines in (a) and (b) indicate averaged concentration during 1910 to 2013.



**Table 4.** Average number concentrations (#/mL) from 1910 to 1950, from 1950 to 1990, and from 1990 to 2013.

|           | 0.70–2.01 μm | 2.01–6.49 μm |
|-----------|--------------|--------------|
| 1910−1920 | 16133.0      | 2010.1       |
| 1910−1950 | 16026.7      | 2203.3       |
| 1950−1990 | 17205.3      | 2390.6       |

The coarse/fine particle ratio obtained from the Coulter counter measurements also indicate an increase in the local mineral dust supply to the EGRIP site. The Coulter counter dust profiles (Table 4, Fig. 12) show that the average number concentration for particles with diameters larger than 0.70 μm and 2.01 μm are 7.4% and 8.5% higher in the 1990–2013 samples than in the 1950–1990 samples, respectively. Particles with diameters larger than 6.5 μm are omitted as they showed significantly higher standard deviations due to their low concentrations (Fig. A3). The concentration of coarse particles

shows a significant correlation with that of fine particles (r > 0.9), indicating that most coarse particles originate from distant sources that also produce fine particles. However, the coarse (2.01–6.49 μm)/fine (0.70–2.01 μm) particle concentration ratio shows an abrupt increase since 2000, which is negatively correlated with the modelled snow cover fraction anomaly during the summer (June, July, and August) on the southeast coast of Greenland (Fig. 12). The fraction of snow cover is expected to be directly related to the duration of snow cover. Given the reduction in the fraction of snow cover for the east coast of

Greenland in recent warming periods, it is likely that the snow cover duration was also reduced, which may have caused an increase in local dust emissions. The slightly higher proportion of Type C minerals from 1990 to 2013 (Fig. 9) may support this argument. Enhanced transport of local Greenland dust after the 1980s has also been reported mainly at the coastal Greenland ice sheet, including the SIGMA-D ice core (Simonsen et al., 2019; Amino et al., 2021; Nagatsuka et al., 2021), and even at the inland ice sheet (Kjær et al., 2022). On the other hand, the increase in the coarse/fine particle concentration

ratio during the 1960s is not consistent with the snow cover fraction anomaly, but is negatively correlated with the NAO index. NAO is strongly correlated with the intensity and incidence of high-pressure blocking over Greenland (Woollings et al., 2010; Hanna et al., 2014) and can change temperature, atmospheric circulation patterns and the ice-core dust transportation processes. The back-trajectory analysis showed that the contribution from Northern Eurasia was significantly low in the mid-1960s, which could have caused a relatively large contribution of local dust and hence a higher proportion of

coarse particles. We suggest that the increase in the coarse particle concentration in the 1960s was likely affected by changes in atmospheric transport.

### 4.3 Volcanic particles

Studies have suggested that Greenland ice cores preserve records of past volcanic events in the Northern Hemisphere, identified as volcanic acid and ash (tephra) layers. Chemical and physical profiles of ice and volcanic dust from Greenland

ice cores show a continuous record of volcanism over the past 110,000 years and indicate that the frequency of volcanism was higher during the last glacial-interglacial transition and early Holocene (e.g., Zielinski et al., 1997; Cook et al., 2022). Several volcanic events have also been identified in recent years (e.g., Lyons et al., 1990; Crowley et al., 1993; Sigl et al., 2013). The nss-$SO_4^{2-}$ concentration records from the North Greenland Eemian Ice Drilling (NEEM) northwestern Greenland ice core identified eight volcanic eruptions in the 1900s (Sigl et al., 2013). Moreover, acidity and conductivity records for

shallow Greenland ice cores show more than ten volcanic events since 1970 (Kjær et al., 2022).

The $SO_4$ profiles for the EGRIP ice core also likely indicate several volcanic eruptions during the past 100 years, which have been identified in the NEEM and shallow firn EGRIP cores (Sigl et al., 2013; Kjær et al., 2022) (Fig. A1). However, in the SEM-EDS analysis, the EGRIP ice-core minerals exhibited characteristics of volcanic ash only for the 1910–1920 sample



(Table 3, Fig. 6), which had a relatively high Si content (70–80%) along with Al, Na, and K (10% each). This composition likely corresponds to minerals originating from rhyolite. The world's largest 20th-century volcanic eruption, that of the Katmai volcano on the Alaska Peninsula, occurred in 1912. The eruption ejected mainly high-silica rhyolite with dacite and andesite (Hildreth and Fierstein, 2000), and thus the ice-core volcanic dust likely originated from this eruption.

Because of the high altitude and remoteness of the polar ice sheet, only material from major volcanic eruptions is deposited in Greenland ice cores (Svensson et al., 2000). The Katmai eruption in 1912 was the only eruption with a volcanic explosivity index (VEI) of 6 at the high latitudes of the Northern Hemisphere in the 20th century. This volcanic event has been confirmed by high electrical conductivity (obtained using a dielectric profiling system) and $SO_4^{2-}$ and $Cl^-$ signals from central and north Greenland ice cores (Lyons et al., 1990; Coulter et al., 2012; Vallelonga et al., 2014; Cole-Dai et al., 2018; Kurosaki et al., 2020; Nagatsuka et al., 2021). Another VEI-6 eruption, namely the Mount Pinatubo (Indonesia) eruption in 1991, and two VEI-5 eruptions, namely the Mount St. Helens (U.S.) eruption in 1980 and the Mount Agung (Indonesia) eruption in 1963, have also been identified in the northwestern SIGMA-A and NEEM ice cores (Sigl et al., 2013; Kurosaki et al., 2020). In addition to these major events, Kjær et al. (2022) identified some VEI-2 to VEI-4 eruptions, likely originating from Iceland, U.S., Russia, and/or Mexico in a shallow firn EGRIP core from 1971 to 2011. However, SEM observations did not find volcanic dust from eruptions other than the Katmai eruption in our EGRIP ice core. This is likely due to the 10-year sample resolution of the ice-core dust, which makes it difficult to find smaller amounts of volcanic ash from minerals, and the large spatial variability of volcanic deposition on the ice sheet, as reported in previous studies (e.g., Gao et al., 2008; Kjær et al. 2022). No strong evidence of the 1991 Pinatubo eruption was found in the Northeast Greenland Ice Stream (NEGIS) ice core (Vallelonga et al., 2014), and evidence was found in only one of two shallow firn cores (Kjær et al., 2022), both of which were drilled close to the EGRIP site. The redistribution of snow on the ice sheet surface may affect the nonuniform deposition of volcanic dust, even within a small area.

## 5. Conclusions

We presented a new high-temporal-resolution ice-core record of dust mineralogy obtained from northeastern Greenland over the past 100 years. We studied regional variations in the mineralogy of dust through a comparison with the ice-core record from northwestern Greenland. Our SEM-EDS findings provide morphological and mineralogical properties of individual dust particles in the EGRIP ice core, which revealed that the ice core dust comprised mainly of silicate minerals, namely quartz, mafic minerals, feldspars, and clay minerals, including kaolinite, illite, smectite, micas, and chlorite. The diameter of most particles was smaller than 2 µm, which is smaller than that for the SIGMA-D ice core, implying that the EGRIP ice core contained dust particles that were transported over longer distances. The EGRIP ice core showed lower compositional variation among the samples compared to that for the SIGMA-D ice core, indicating that the mineral dust sources did not change dramatically during the period. SIGMA-D ice-core minerals varied on a multi-decadal scale, which is strongly related to temperature changes in Greenland. This was caused by an increasing contribution of minerals originating from local ice-free areas because of shorter snow/ice cover duration in the Greenland coastal region during the warming periods. Therefore, the subtle variations in the EGRIP mineral composition are likely due to a smaller contribution of local dust.

The silicate mineral composition differed greatly between the two ice cores; micas and chlorite, which generally form in high-latitude areas, were abundant in the EGRIP ice core, whereas kaolinite, which likely originated from northern Canada, was abundant in the SIGMA-D ice core. This indicates that the EGRIP ice-core dust likely originated from different geological sources than those for the SIGMA-D dust. The back-trajectory results showed that the contribution to the air mass that arrived at the EGRIP site was smaller from North America and larger from Northern Eurasia compared to that for the SIGMA-D ice core, corresponding to a higher relative abundance of mineral types in the EGRIP ice core than in the SIGMA-D ice core. The higher illite content in the EGRIP ice core also suggests dust transportation from Asian deserts. Although most of the air mass that arrived at the EGRIP site came from the Greenland coast, as shown in the trajectory



analysis, the mineral particle size and composition results showed that the local dust contribution was likely small. We concluded that the EGRIP ice-core dust was supplied mainly from Northern Eurasia, as well as from North America, and Asia.

The $SO_4$ profiles for the EGRIP ice core indicated several volcanic eruptions during the past 100 years. However, the SEM-
EDS analysis identified ice-core minerals that exhibited the characteristics of volcanic ash only for a sample from 1910–1920 with a composition that likely corresponds to rhyolitic minerals. The ice-core volcanic dust likely originated from the Katmai eruption on the Alaska Peninsula in 1912, the world's largest 20th-century eruption.

Although further analyses are necessary to identify detailed source areas for the EGRIP ice-core minerals and their mixing ratios, our study provides a better understanding of the spatial and temporal variations in mineral dust sources within the
Greenland Ice Sheet.





**Appendix A: Figures**

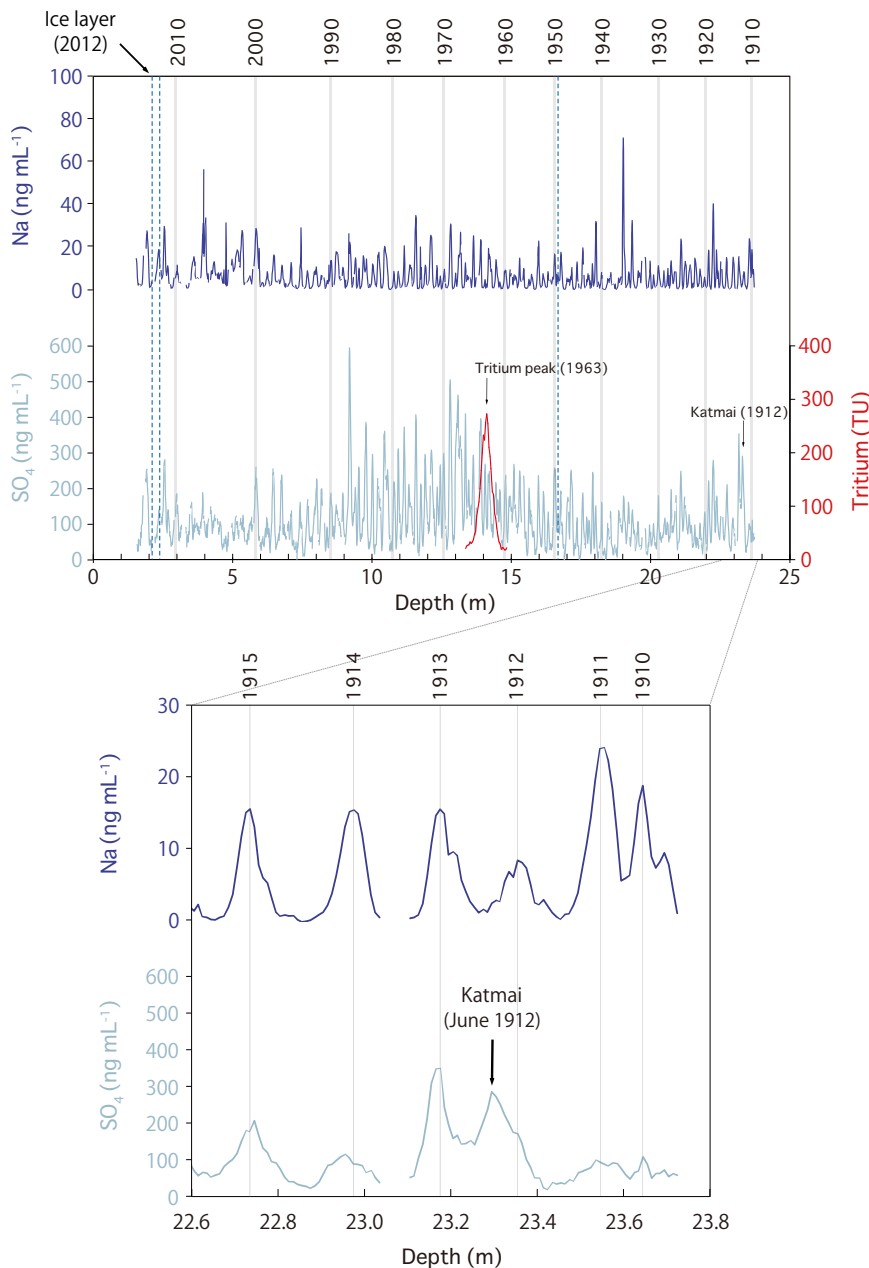

**Figure A1.** Na, SO₄, tritium, and ice layer records in upper 23.64 m (1910–2013) of EGRIP ice core (top). Major volcanic signals identified in SO₄ record are shown. Enlarged record from 22.6 to 23.725 m (bottom).






**Figure A2.** (a) Comparison of size distribution and log-normal fitting results (mode: mode diameter; $R^2$: half peak width) for EGRIP ice core mineral dust among samples. (b) Historical changes in mode diameter and $R^2$.





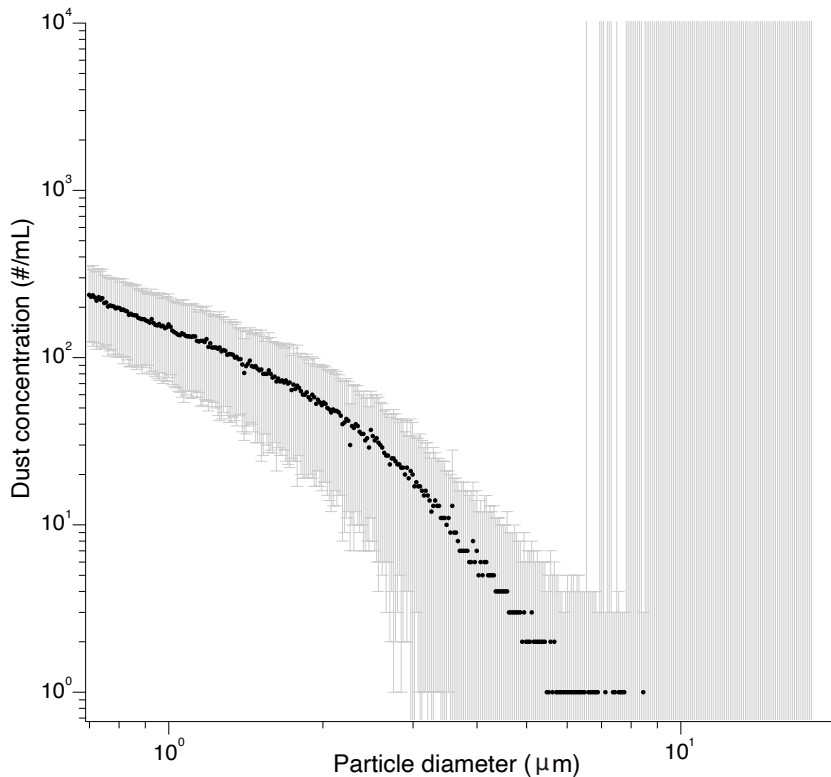

**Figure A3.** Average number concentration of EGRIP ice core dust with diameters of 0.70–18.00 μm. The error bars indicate the standard deviation.



*Data availability*

MIROC6 model output prepared for CMIP6 LS3MIP experiments is available at the following references:

https://doi.org/10.22033/ESGF/CMIP6.5622 (land-hist: Onuma and Kim, 2020a),

https://doi.org/10.22033/ESGF/CMIP6.5627 (land-hist-cruNcep: Onuma and Kim, 2020b),

https://doi.org/10.22033/ESGF/CMIP6.5628 (land-hist-princeton: Onuma and Kim, 2020c), and

https://doi.org/10.22033/ESGF/CMIP6.5629 (land-hist-wfdei: Onuma and Kim, 2020d).

Data on $\delta^{18}O$ and concentrations of sodium, sulfate (Na and $SO_4$), and tritium will be submitted to the ADS (Arctic Data
archive System) database for public use in further analysis.

*Author contributions*

NN designed the study and carried out the ice core dust analysis and wrote the manuscript with the help of KGA, KoF, and
FN. TJP and DDJ drilled the ice core. YK, MH, JO, KaF, YOT, KK, AY, and KGA conducted the CFA analysis and data
processing. YK, FN, and KGA conducted the Coulter counter analysis. YK, KGA, FN, SOR, and GS determined the
chronology of the ice core and YK also wrote the related paragraphs. NK conducted the tritium measurement and wrote the
related paragraphs. KoF conducted the back-trajectory analysis. YO conducted the CMIP6 model analysis. DDJ led the
EGRIP project. All authors discussed and commented on the paper.

*Competing interests*
The authors declare that they have no conflict of interest.

*Acknowledgements*

We would thank to members of the EGRIP projects for their generously support. EGRIP is directed and organized by the
Centre for Ice and Climate at the Niels Bohr Institute and US NSF, Office of Polar Programs. It is supported by funding
agencies and institutions in Denmark (A. P. Møller Foundation, UCPH), US (US NSF, Office of Polar Programs), Germany
(AWI), Japan (NIPR and ArCS), Norway (BFS), Switzerland (SNF), France (IPEV, IGE), and China (CAS). We also would
like to thank Hiromi Okumura at NIPR for supporting the Coulter counter analysis. This research has been supported by the
JSPS KAKENHI Grant Numbers JP18H04140, JP19K20443, JP20K19962, JP22J40168, the Arctic Challenge for
Sustainability (ArCS) Project (Program Grant Number JPMXD130000000)), the Arctic Challenge for Sustainability II
(ArCS II) Project (Program Grant Number JPMXD1420318865), the Integrated Research Program for Advancing Climate
Models (Program Grant Number JPMXD0717935457), the Advanced Studies of Climate Change Projection (Program Grant
Number JPMXD0722680395), the Environment Research and Technology Development Funds (JPMEERF20172003 and
JPMEERF20202003) of the Environmental Restoration and Conservation Agency of Japan.

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
