# Peer review of "Regional variations in mineralogy of dust in ice cores obtained from northeastern and northwestern Greenland over the past 100 years"

_EGUsphere, 2023_

## Author Comment (AC1)

We thank very much the anonymous referee #1 for taking the time to review our manuscript and giving helpful suggestions and comments. We here respond to the referee #1's comments:

**[General comments from referee #1]**

In this manuscript Nagatsuka and colleagues analyze the mineral content of dust particles in a shallow ice core from central Greenland, and estimate the potential source contribution through backtrack trajectory modeling. They compare their results with a similar core drilled further west and published previously.

The main contribution of this manuscript are the detailed mineralogical analyses of this new shallow core since 1910. Although the results are new, they are rather incremental and it is not clear how these new data are improving our knowledge of Central Greenland dust advection or source contribution. That dust in central Greenland mostly originates from distant sources (mostly in East Asia) and not local ones was already known from other cores. This study mostly repeats this result at higher resolution. In addition, the authors imply links between their results with recent warming in Greenland, which is poorly supported since any kind of analyses including Atlantic and Pacific oscillations are missing. Finally, the discussion of volcanic particles is mostly a literature review without any contribution from this manuscript.

For these reasons I suggest to reject this manuscript as it does not include sufficient scientific advances for Climate of the Past. Instead, I suggest to publish these results in a more specialized journal.

**[Author response]**

We agree with your suggestions that we should discuss the link between mineral composition records and Atlantic and Pacific oscillations in much more details and that we should compare the results not only with a northwest Greenland ice core (SIGMA-D), but also with other central, east, and southeast Greenland. However, we would like to argue that this study demonstrates the following three important results for the first time that improve our knowledge of global climate and environment changes including Greenland.

1. We have known that mineral dust on the inland Greenland ice sheet primarily originates from distant sources, as suggested by referee #1. However, previous studies mainly investigated ice core dust sources from glacial periods using mainly the Sr-Nd-Pb (Hf) isotopes (e.g., Biscaye et al., 1997; Svensson et al., 2000; Lupker et al., 2010; Újvári et al., 2022). Little is known about the dust composition and sources from interglacial periods, especially in the modern times, when the dust concentrations are significantly lower than the glacial periods,

mainly due to the large sample amounts required for Sr-Nd-Pb (Hf) analyses. Furthermore, recent studies have revealed that the ability of ice nucleating particles (INP), which alter the cloud microphysics and lifetime in the Arctic, differs depending on the mineral types (e.g., Koop and Mahowald., 2013; Tobo et al., 2019). Thus, it is crucial to identify the variations in the composition and morphology of mineral dust in recent years.

This study is the first to demonstrate a high-temporal resolution record of the composition and morphology of mineral dust on inland ice sheet, which are not only important to understand modern INP variations but also reveals sources in a central Greenland (EGRIP) ice core over the past 100 years when global warming is remarkably progressing. Although our conclusion that one of the major sources of the EGRIP ice core dust is distant deserts may not sound new, it is very new and valuable information that there are no significant differences in dust sources between glacial and present periods since only a few study has confirmed it. This result is different from the results obtained from Antarctic ice cores, which show different source contributions between interglacial and glacial periods. However, we didn't stress this enough in the manuscript and gave the impression that our result is not new. When we revise our manuscript, we will stress this point.

Some studies have also revealed temporal variations of the Greenland ice core dust sources in modern days. For example, Amino et al (2021) and Kjær et al (2022) showed records of dust concentrations and fluxes from southeast and central Greenland ice cores respectively, and revealed a recent increase in dust sources from local ice-free areas. However, their records go back only to 1960s. Furthermore, they did not discuss contributions of dust from other source areas. Bory et al (2003) analyzed the Sr-Nd isotopes, and Drab et al., (2002) and Donarummo et al. (2003) carried out the SEM-EDS analysis of dust from Greenland snow and ice cores. The authors revealed dust source records in the 1900s, but they cover just for a few years. Thus, we give the first results showing temporal variations in ice core dust composition and sources from multiple regions covering such a long period (100 years).

2.  The source of mineral dust on inland ice sheets has been generally thought to be primarily distant deserts. However, we demonstrate a possibility of a recent increase in local dust contribution to the inland ice sheet based on our high-temporal resolution records of ice core mineral size and composition and snow cover fraction anomaly. The possibility of local dust increase has also been suggested by Kjær et al (2022). Our result is considered valuable since there is still very little knowledge of local dust contribution to inland ice sheets.

    However, our study has missed a discussion related to Atlantic and Pacific oscillations, as suggested by the referee #1. We will carry out analyses related to them and discuss in our revised manuscript about both possibilities of local dust increase and the oscillations as causes

for our mineral composition records. If the oscillations turn out to be more likely cause, it also supports the importance of our study because we can find such a result only because of our high-temporal resolution analysis.

3.  This study is the first to demonstrate the regional variation in Greenland dust sources over the 100 years by comparing inland and coastal ice cores. It is essential to identify spatial variations in the Greenland ice core dust source records, which would be a key to understand differences in climate and environment conditions within Greenland and atmospheric circulation patterns.

For these reasons, we believe that our study provides new perspectives on reconstruction of the past climate/environment and fluctuations of the Greenland ice sheet, thus contributing to scientific advances for Climate of the Past and earth sciences. However, the current version of the manuscript does not clearly show the significance of our study. We would like to revise it and emphasize these points.

**[Major comments from referee #1]**
The authors mostly compare their results with an ice core from northeast Greenland (sigma-d) for which similar data are available. However, the comparison to central Greenland, east Greenland (Renland) and southeast Greenland (Dye-3) should be included in the discussion. In particular, the comparison with NGRIP should be made, as the claim that EGRIP represents Eastern Greenland and NGRIP central Greenland is a bit shaky, considering both sites are at similar altitudes and quite close to each other.

The authors group Europe and NorthEast Asia, as well as Africa and SouthEast Asia into single potential source areas in their analysis. Considering the long debate about Asian, European and African dust sources for Greenland, these should probably be split into four, unless the authors can justify their choice.

The authors talk about trends in the data in various sections of the manuscript, in particular comparing the last 20 years with the mid-section of the core. In particular, the authors imply that the recent warming has been responsible for various changes in dust mineralogy and concentrations. But looking at the complete record, these look more like multidecadal oscillations to me and should not be described as trends. Although recent warming in Greenland may have been responsible for some of the observed changes, such a hypothesis has to be put in context with the complete oscillations shown in the records. The authors briefly mention NAO at some

point, but the link between their record and various Atlantic and Pacific oscillations should be discussed in much more details.

**[Author response]**

We will compare our results with those obtained from other Greenland ice cores as suggested by referee #1. However, most of the previous studies have analyzed the dust from the last glacial period. Furthermore, there are no continuous records of size, composition, and source data of ice core minerals for the past 100 years. Thus, the results may not be directly comparable.

We recalculated the back-trajectories as follows. Potential dust source (land) areas are divided into eight regions following referee #1's suggestion; the Greenland Ice Sheet (GrIS), the Greenland coast (GrC), North America (NA), Europe (EU), Russia (RUS), Central Asia (CA), Southeast Asia (SA), Middle East (ME), and Africa (AF) (Fig. Rep1). Considering the dust sources, the ocean and GrIS were excluded from the calculation.
Revised version of figs 2 and 7 are shown in Figs. Reply 1 and 2, respectively. We will replace them in our revised manuscript.

We will carry out analyses related to Atlantic and Pacific oscillations as described above.

**[Minor Comments from referee #1]**

Line 18: Abstract could benefit from a more general introductory phrase at the beginning.

Line 33-34: Since Greenland is an island, all air masses must come from a coast. Be more precise.

Line 38: 100,000 years seems a bit short for the geological timescale, although I am not a geologist and may be wrong. Maybe Milankovitch timescale?

Line 42: "ice-core dust shows…". Also this is only shown for Central Greenland, not the whole of Greenland.

Line 44-45: This is not at all the message of Svensson et al., 2000. Generally, I very much doubt that the seasonal variability in dust advection to Greenland is due to climate change…

Line 46: Not "predict". "estimate" maybe.

Line 47: "partly responsible". Grain size and partial melt is very important as well for albedo.

Line 53-58: The message of these phrases is unclear. Are you suggesting to collect dust from outcropped ice in the ablation zone to measure old dust? Or just dust in fresh snow on the surface? Then why talk about the movement of ice and dust through the ice sheet?

Line 60: Ujvari et al. used Hf not Pb.

Line 75: Can you give some references to support that hypothesis?

Line 117: Is the Beckman CC located in a normal laboratory or a clean room or a laminar flow bench? What kind of aperture tube was used?

Line 145, Table 1: Why is South America included as a possible source for Type A particles? I'm not saying it's wrong (although I do doubt it), but I wonder why it was included in the list.

Lines 171-174: Snow cover fractions vary substantially from one model to another. Please provide and uncertainty estimate due to the choice of the model.

Lines 209-216: What do the numerical ranges indicate? 1-sigma range? If so how were the mean and standard deviations calculated?

**[Author response]**
All the minor comments and suggestions raised by the referee will be respected in our revised manuscript.

[Figure]

Figure reply1 (revised from fig. 2): Map showing (a) location of EGRIP and SIGMA-D ice core sites in Greenland and five regions used for calculating regional contribution (GrIS: Greenland Ice Sheet, grey; GrC: Greenland coast, purple; NA: North America, orange; EU: Europe, blue; RUS: Russia, light blue; CA: Central Asia, yellow; SA: Southeast Asia, green; ME: Middle East, red; and AF: Africa, brown) and probability distribution for air mass at (b) EGRIP and (c) SIGMA-D sites from 7-day three-dimensional back trajectory analysis from 1958 to 2014.

[Figure]

Figure reply 2 (revised from fig. 7): Annual variations in regional contribution of air mass to (a) EGRIP and (b) SIGMA-D sites excluding ice sheet and ocean areas. GrC, NA, EU, RUS, CA, SA, ME and AF denote the ice-free Greenland coastal region (Fig. 1), North America, Europe, Russia, Central Asia, Southeast Asia, Middle East, amd Africa, respectively (Fig. 2a).

References:

Amino, T., Iizuka, Y., Matoba, S., Shimada, R., Oshima, N., Suzuki, T., Ando, T., Aoki, T., and Fujita, K. (2021) Increasing dust emission from ice free terrain in southeastern Greenland since 2000, Polar Sci., 27, 100599.

Biscaye, P. E., Grousset, F. E., Revel, M., Van der Gaast, S., Zielinski, G. A., Vaars, A., and Kukla, G. (1997) Asian provenance of glacial dust (stage 2) in the Greenland Ice Sheet Project 2 ice core, Summit, Greenland, J. Geophys. Res., 102, 26765–26781.

Bory, A. J.-M., Biscaye, P. E., and Grousset, F. E. (2003) Two distinct seasonal Asian source regions for mineral dust deposited in Greenland (NorthGRIP), Geophys. Res. Lett., 30, 1167.

Donarummo, J., Ram, M., and Stoermer, E. F. (2003) Possible deposit of soil dust from the 1930's U.S. dust bowl identified in Greenland ice, Geophys. Res. Lett., 30, 1269.

Drab, E., Gaudichet, A., and Jaffrezo, J. L. (2002) Mineral particles content in recent snow at Summit (Greenland). Atmospheric Environment., 36, 5365–5367.

Kjær, H. A., Zens, P., Black, S., Lund, K. H., Svensson, A., and Vallelonga, P. (2022) Canadian forest fires, Icelandic volcanoes and increased local dust observed in six shallow Greenland firn cores, Clim. Past, 18, 2211–2230.

Koop, T., and Mahowald, N. (2013) The seeds of ice in clouds. *Nature* **498**, 302–303.

Lupker, M., Aciego, S. M., Bourdon, B., Schwander, J., and Stocker, T. F. (2010) Isotopic tracing (Sr, Nd, U and Hf) of continental and marine aerosols in an 18th century section of the Dye-3 ice core (Greenland), Earth Planet. Sci. Lett., 295, 277–286.

Svensson, A., Biscaye, P. E., and Grousset, F. E. (2000) Characterization of late glacial continental dust in the greenland ice core project ice core, J. Geophys. Res., 105, 4637–4656.

Újvári, G., Klötzli, U., Stevens, T., Svensson, A., Ludwig, P., Vennemann, T., Gier, S., Horschinegg, M., Palcsu, L., Hippler, D. Kovács, J., Di Biagio, C. and Formenti, P. (2022) Greenland ice core record of last glacial dust sources and atmospheric circulation. J. Geophys. Res. Atmo.s, 127, e2022JD036597.

---

## Author Comment (AC2)

*Reply to Referee#3*

December 19, 2023

Dear Referee #3,

We greatly appreciate a number of valuable comments on our manuscript entitled "Regional variations in mineralogy of dust in ice cores obtained from northeastern and northwestern Greenland over the past 100 years" by Nagatsuka et al. submitted to the journal Climate of the Past. Please see enclosed our responses (**blue text**) to each of your comments (**black text**) describing on the following pages.

Best regards,

Naoko Nagatsuka and co-authors
National Insitute of Polar Reserach
E-mail: nagatsuk.naoko@nipr.ac.jp

General statements

Dear authors, the two reviews you already receive present most of my perplexities and doubts about your methodology and robustness of conclusions.

My appreciation is that the paper presents interesting data, and could deserve publication if the treatement was better presented and justified. I only have a few additional suggestions with respect to the points rose by my colleagues.

We would like to thank you for all the valuable comments. According to the suggestions from you and other referees, we will add the results of new analyses and revise our manuscript as below:

Introduction and conclusion:

First, we will clarify and emphasize new and important findings in our study, especially in introduction and conclusion sections. Our SEM-EDS analysis show the following two new results: This study is the first to demonstrate (1) continuous records of a northeast Greenland ice core dust (EGRIP) composition with a high temporal resolution during the last 100 years and (2) their significant differences from northwestern Greenland.

Although previous studies revealed mineral dust sources of the Greenland ice cores, most of the dust were obtained from the last glacial period when the dust concentrations were high. Some studies have analyzed the ice core minerals during the Holocene (8.2-11.6 ka: Han et al., 2018, 4-7 ka: Simonsen et al., 2019, 1997-2001 A.D: Bory et al., 2003a), a period of low dust concentration, however, they needed to concentrate decades to thousands of years of ice for each sample. Thus, our results can provide new and important insight to understand sources and transportation processes of ice core dust as well as climate and environmental change in recent years when global warming is progressing.

Results and Discussion:

We agree that it is difficult to "identity" the sources of the ice core dust accurately only based on our mineralogical and trajectory results as suggested by referee #2. However, our results can clearly show the differences in the dust sources between the EGRIP and SIGMA-D ice core based on the mineral composition. Thus, we will revise the manuscript to emphasize the differences in the two ice core sites and "discuss" possible dust source areas. We will also add some new results of the composition and elemental concentration of the EGRIP ice core dust obtained from the SEM-EDS and ICP-MS analyses to discuss the possible sources and their temporal variations (Figs. Reply. 1 - 4). The details are as follows.

The EGRIP dust showed mineral composition characterized in Asian-sourced dust as described in line 305-307. Thus, Asian desert can be one of possible sources of the dust at EGRIP as the other Greenland ice core sites. Comparing the mineral composition, however, the EGRIP ice core

dust showed significantly lower quartz content and higher illite/micas/chlorite contents than the GRIP ice core dust originated from Chinese deserts (Fig. Reply 1 and 2). We cannot compare the EGRIP mineral composition records directly with those of previous studies because there have been no continuous records of dust composition during the last 100 years. Moreover, the periods analyzed (modern vs mostly glacial), analytical methods (X-ray diffraction analysis (XRD) vs SEM-EDS) and mineral identification methods of the ice core dust were different (XRD used in previous studies cannot separate micas from illite, whereas the elemental peak intensity ratio sorting scheme used in our study cannot separate micas from chlorite completely. We identified them as mixed layer of micas/chlorite). However, the lower quartz abundance may indicate that the EGRIP dust were contributed not only from Asia, but also from other source areas. Thus, the EGRIP ice core dust was originated from multiple sources. Air-mass trajectory results suggests that Northern Eurasia and North America are the best candidate of the other possible sources. Consequently, one of the possible sources of the EGRIP ice core dust is likely to be Asian deserts as the other Greenland ice core dust from glacial periods and the time series of the mineral composition did not change significantly over the last 100 years. These results may not sound exciting or new. But we argue that these results are new and important since no previous study has confirmed it. Although previous studies revealed mineral dust sources of the Greenland ice cores, most of the dust were obtained from the last glacial period when the dust concentrations were high. Furthermore, we also found the new results indicating that the source areas have changed since 1970-1980, which is new insight that could improve the understanding of Greenland ice core dust studies.

[Figure]

Figure Reply 1. Mineral composition of the EGRIP (this study) and GRIP ice cores (Svensson et al., 2000)

[Figure]

Figure Reply 2. Clay mineral composition of the EGRIP (this study) and Hans Tausen, NGRIP, GRIP, Renland, Site A, and Dye-3 ice cores (Bory et al., 2003).

Although mineral composition records indicates that the mineral sources of the EGRIP dust have not changed dramatically over the past 100 years as described in Line 26, the ternary clay mineralogy diagram (Fig. 10) and the elemental concentration ratios of the EGRIP ice core dust analyzed by the SEM-EDS and ICP-MS, respectively, suggest that there is slight difference in the dust sources in different periods. In ternary clay mineralogy diagram, the dust from 1910 to 1970 were plotted close to glacial dust from the GRIP ice core as well as the Asian dust, whereas the dust from 1970 to 2013 were not. Furthermore, the samples from 1980 to 2013 showed the steeper slope in scatter plots of elemental concentrations analyzed by ICP-MS (Fig. Reply 3). These results indicate that the contribution rate from the possible sources has changed since 1970-1980, which is new insight that could improve the understanding of Greenland ice core dust.

One of the possible causes of this change is decreasing Asian dust supply to the EGRIP. Although the number of the particles may not be sufficient to show accurate percentages, the illite content of the EGRIP showed decreasing trend since 1980 that is similar to that for dust occurrence in Asian deserts (e.g. Zhang et al., 2020). This is likely due to global warming that could have led to a decrease in atmospheric temperature gradients and decline in wind speed and then decreasing dust storm frequency/intensity. We also compared the nss Ca/Al ratios with average volume of dust particles calculated from volume concentration divided by number concentration of the EGRIP ice core, which can reflect the contribution of evaporite minerals that are abundant in deserts (Formenti et al., 2011) and the distance from the source areas, respectively (Fig. Reply 4). The nss Ca/Al ratios were slightly lower but average volume of dust particles were higher from 1970 to 1990. These results indicate that the contribution from distant deserts likely decreased during the period, which are consistent with the hypothesis that the dust contribution has changed since 1970-1980.

The higher Ca/Al ratios and lower average volume of dust particles of the EGRIP dust after 1990 imply the possibilities that the dust has originated from distant sources with higher salinity environments in this period. Previous studies have revealed that there are large variations in the Ca/Al ratios among Asian deserts and that the Taklamakan deserts has higher values than the other deserts (e.g. Formenti et al., 2011). The EGRIP ice core dust might be derived from such sources.

We will add some sentences in the manuscript to describe these points.

[Figure]

Figure Reply 3. Elemental concentration ratios of microparticles in the EGRIP ice core anayzed by ICP-MS.

[Figure]

Figure Reply 4. Historical records of elemental concentration ratios ((a) Si/Al, (b) nssK/Al, (c) Fe/Al, (d) nssMg/Al, (e) nssCa/Al, and (f) average volume of dust particles in the EGRIP ice core.

Detailed comments

- is the X-ray self attenuation for light elements taken into account for the EDS analysis? this can be severe for elements such as Al when large particles are present and could false the mineralogical attribution

Yes. We consider X-ray self-attenuation using the "ZAF correction method". The ZAF correction involves accounting for three effects on characteristic X-ray intensity during quantitative analysis: 1) the atomic number (Z) effect, 2) the absorption (A) effect, and 3) the fluorescence excitation (F) effect.

- there is too little description of the way by which the size distributions are obtained

We will add the description of how to obtain the size distribution data in the revised manuscript.

- I would suggest to present the size distribution as number of particles per unit size class and use a log scale for the x-axis

We will present figures of size distribution (Figs 4 and A2) in the description way you suggested in the revised manuscript.

- the calcium depletion could be due to size sorting and not solubility, have you considered that?

Many thanks for the suggestion. We haven't considered the possibility of the depletion and will describe it in the revised manuscript.

- back trajectories are only showing provenance but need to be complemented by precipitation fields if you want to obtain deposition. Dry deposition can be approximated (roughly) by gravitational settling if you know the size distribution.

Does this comment mean "consider the removal effect by precipitation and by particle size dependency along the air mass pathways"? If our understanding is correct, it is difficult. The code for analysis used in the trajectory analysis can consider the precipitation only at the targeted site. We emphasize that the trajectory results show the differences of dust sources between the EGRIP and SIGMA-D site. We will not "identify" but "discuss" possible dust source areas in the revised manuscript. We think that our trajectory results are sufficient to support the differences in dust sources between the two ice core sites.

References:

Bory, A. J.-M., Biscaye, P. E., Piotrowski, A. M., and Steffensen, J. P.: Regional variability of ice core dust composition and provenance in Greenland, Geochem. Geophys. Geosyst., 4, 1107, https://doi.org/10.1029/2003GC000627, 2003.

Formenti, P., Schütz, L., Balkanski, Y., Desboeufs, K., Ebert, M., Kandler, K., Petzold, A., Scheuvens, D., Weinbruch, S., and Zhang, D.: Recent progress in understanding physical and chemical properties of African and Asian mineral dust, Atmos. Chem. Phys., 11, 8231–8256, https://doi.org/10.5194/acp-11-8231-2011, 2011.

Han, C., Hur, S. D., Han, Y., Lee, K. Hong, S., Erhardt, T., Fischer, H., Svensson, A. M., Steffensen, J. P., Vallelonga, P.: High-resolution isotopic evidence for a potential Saharan provenance of Greenland glacial dust, Sci. Rep., 8, 15582, | https://doi.org/10.1038/s41598-018-33859-0 3, 2018.

Maggi, V.: Mineralogy of atmospheric microparticles deposited along the Greenland Ice Core Project ice core, J. Geophys. Res., 102, 26725–26734, https://doi.org/10.1029/97JC00613, 1997.

Simonsen, M. F., Baccolo, G., Blunier, T., Borunda, A., Delmonte, B., Frei, R., Goldstein, S., Grinsted, A., Kjær, H. A., Sowers, T., Svensson, A., Vinther, B., Vladimirova, D., Winckler, G., Winstrup, M., and Vallelonga, P.: East Greenland ice core dust record reveals timing of Greenland ice sheet advance and retreat, Nat. Commun., 10, 4494, https://doi.org/10.1038/s41467-019-12546-2, 2019.

Svensson, A., Biscaye, P. E., and Grousset, F. E.: Characterization of late glacial continental dust in the greenland ice core project ice core, J. Geophys. Res., 105, 4637–4656, https://doi.org/10.1029/1999JD901093, 2000.

Újvári, G., Klötzli, U., Stevens, T., Svensson, A., Ludwig, P., Vennemann, T., Gier, S., Horschinegg, M., Palcsu, L., Hippler, D. Kovács, J., Di Biagio, C. and Formenti, P.: Greenland ice core record of last glacial dust sources and atmospheric circulation. J. Geophys. Res. Atmo.s, 127, e2022JD036597, https://doi.org/10.1029/2022JD036597, 2022.

Zhang, Y., Mahowald, N., Scanza, R. A., Journet, E., Desboeufs, K., Albani, S., Kok, J. F., Zhuang, G., Chen, Y., Cohen, D. D., Paytan, A., Patey, M. D., Achterberg, E. P., Engelbrecht, J. P., and Fomba, K. W.: Modeling the global emission, transport and deposition of trace elements associated with mineral dust, Biogeosciences, 12, 5771–5792, https://doi.org/10.5194/bg-12-5771-2015, 2015.

---

## Author Comment (AC3)

*Reply to Referee#1*

December 19, 2023

Dear Referee #1,

We greatly appreciate a number of valuable comments on our manuscript entitled "Regional variations in mineralogy of dust in ice cores obtained from northeastern and northwestern Greenland over the past 100 years" by Nagatsuka et al. submitted to the journal Climate of the Past. Please see enclosed our responses (**blue text**) to each of your comments (**black text**) describing on the following pages.

Best regards,

Naoko Nagatsuka and co-authors
National Insitute of Polar Reserach
E-mail: nagatsuka.naoko@nipr.ac.jp

General statements

In this manuscript Nagatsuka and colleagues analyze the mineral content of dust particles in a shallow ice core from central Greenland, and estimate the potential source contribution through backtrack trajectory modeling. They compare their results with a similar core drilled further west and published previously.

The main contribution of this manuscript are the detailed mineralogical analyses of this new shallow core since 1910. Although the results are new, they are rather incremental and it is not clear how these new data are improving our knowledge of Central Greenland dust advection or source contribution. That dust in central Greenland mostly originates from distant sources (mostly in East Asia) and not local ones was already known from other cores. This study mostly repeats this result at higher resolution. In addition, the authors imply links between their results with recent warming in Greenland, which is poorly supported since any kind of analyses including Atlantic and Pacific oscillations are missing. Finally, the discussion of volcanic particles is mostly a literature review without any contribution from this manuscript.

For these reasons I suggest to reject this manuscript as it does not include sufficient scientific advances for Climate of the Past. Instead, I suggest to publish these results in a more specialized journal.

We would like to thank you for all the valuable comments. We agree with your suggestions that we should discuss the link between mineral composition records and Atlantic and Pacific oscillations in much more detail. Thus, we will compare the variations in mineral composition with those in such oscillations in the revised manuscript. We also agree with the suggestion to compare the results not only with a northwest Greenland ice core (SIGMA-D) but also with other cores in Greenland. Therefore, we compared the mineral composition of the EGRIP ice core dust with that of the GRIP ice core dust. (Fig. Reply 1).

Consequently, one of the possible sources of the EGRIP ice core dust is likely to be Asian deserts as the other Greenland ice core dust from glacial periods and the time series of the mineral composition did not change significantly over the last 100 years. These results may not sound exciting or new, as suggested by you and referee #2. But we argue that these results are new and important since no previous study has confirmed it. Although previous studies revealed mineral dust sources of the Greenland ice cores, most of the dust were obtained from the last glacial period when the dust concentrations were high. Furthermore, we also found the new results indicating that the source areas have changed since 1970-1980, which is new insight that could improve the understanding of Greenland ice core dust. According to the suggestions from you and other referees, we will add the results of new analyses and revise our manuscript as below:

Introduction and conclusion:

First, we will clarify and emphasize new and important findings in our study, especially in introduction and conclusion sections. Our SEM-EDS analysis show the following two new results: This study is the first to demonstrate (1) continuous records of a northeast Greenland ice core dust (EGRIP) composition with a high temporal resolution during the last 100 years and (2) their significant differences from northwestern Greenland.

Although previous studies revealed mineral dust sources of the Greenland ice cores, most of the dust were obtained from the last glacial period when the dust concentrations were high. Some studies have analyzed the ice core minerals during the Holocene (8.2-11.6 ka: Han et al., 2018, 4-7 ka: Simonsen et al., 2019, 1997-2001 A.D: Bory et al., 2003a), a period of low dust concentration, however, they needed to concentrate decades to thousands of years of ice for each sample. Thus, our results can provide new and important insight to understand sources and transportation processes of ice core dust as well as climate and environmental change in recent years when global warming is progressing.

Results and Discussion:

We agree that it is difficult to "identity" the sources of the ice core dust accurately only based on our mineralogical and trajectory results as suggested by referee #2. However, our results can clearly show the differences in the dust sources between the EGRIP and SIGMA-D ice core based on the mineral composition. Thus, we will revise the manuscript to emphasize the differences in the two ice core sites and "discuss" possible dust source areas. We will also add some new results of the composition and elemental concentration of the EGRIP ice core dust obtained from the SEM-EDS and ICP-MS analyses to discuss the possible sources and their temporal variations (Figs. Reply. 1 - 4). The details are as follows.

The EGRIP dust showed mineral composition characterized in Asian-sourced dust as described in line 305-307. Thus, Asian desert can be one of possible sources of the dust at EGRIP as the other Greenland ice core sites. Comparing the mineral composition, however, the EGRIP ice core dust showed significantly lower quartz content and higher illite/micas/chlorite contents than the GRIP ice core dust originated from Chinese deserts (Fig. Reply 1 and 2). We cannot compare the EGRIP mineral composition records directly with those of previous studies because there have been no continuous records of dust composition during the last 100 years. Moreover, the periods analyzed (modern vs mostly glacial), analytical methods (X-ray diffraction analysis (XRD) vs SEM-EDS) and mineral identification methods of the ice core dust were different (XRD used in previous studies cannot separate micas from illite, whereas the elemental peak intensity ratio sorting scheme used in our study cannot separate micas from chlorite completely. We identified them as mixed layer of micas/chlorite). However, the lower quartz abundance may indicate that

the EGRIP dust were contributed not only from Asia, but also from other source areas. Thus, the EGRIP ice core dust was originated from multiple sources. Air mass trajectory results suggests that Northern Eurasia and North America are the best candidate of the other possible sources.

[Figure]

Figure Reply 1. Mineral composition of the EGRIP (this study) and GRIP ice cores (Svensson et al., 2000)

[Figure]

Figure Reply 2. Clay mineral composition of the EGRIP (this study) and Hans Tausen, NGRIP, GRIP, Renland, Site A, and Dye-3 ice cores (Bory et al., 2003).

Although mineral composition records indicates that the mineral sources of the EGRIP dust have not changed dramatically over the past 100 years as described in Line 26, the ternary clay mineralogy diagram (Fig. 10) and the elemental concentration ratios of the EGRIP ice core dust analyzed by the SEM-EDS and ICP-MS, respectively, suggest that there is slight difference in the dust sources in different periods. In ternary clay mineralogy diagram, the dust from 1910 to 1970 were plotted close to glacial dust from the GRIP ice core as well as the Asian dust, whereas the dust from 1970 to 2013 were not. Furthermore, the samples from 1980 to 2013 showed the steeper slope in scatter plots of elemental concentrations analyzed by ICP-MS (Fig. Reply 3). These results indicate that the contribution rate from the possible sources has changed since 1970-1980, which is new insight that could improve the understanding of Greenland ice core dust.

One of the possible causes of this change is decreasing Asian dust supply to the EGRIP. Although the number of the particles may not be sufficient to show accurate percentages, the illite content of the EGRIP showed decreasing trend since 1980 that is similar to that for dust occurrence in Asian deserts (e.g. Zhang et al., 2020). We also compared the nss Ca/Al ratios with average volume of dust particles calculated from volume concentration divided by number concentration of the EGRIP ice core, which can reflect the contribution of evaporite minerals that are abundant in deserts (Formenti et al., 2011) and the distance from the source areas, respectively (Fig. Reply 4). The nss Ca/Al ratios were slightly lower but average volume of dust particles were higher from 1970 to 1990. These results indicate that the contribution from distant deserts likely decreased during the period, which are consistent with the hypothesis that the dust contribution has changed since 1970-1980.

The higher Ca/Al ratios and lower average volume of dust particles of the EGRIP dust after 1990 imply the possibilities that the dust has originated from distant sources with higher salinity environments in this period. Previous studies have revealed that there are large variations in the Ca/Al ratios among Asian deserts and that the Taklamakan deserts has higher values than the other deserts (e.g. Formenti et al., 2011). The EGRIP ice core dust might be derived from such sources.

We will add some sentences in the manuscript to describe these points.

[Figure]

Figure Reply 3. Elemental concentration ratios of microparticles in the EGRIP ice core anayzed by ICP-MS.

[Figure]

Figure Reply 4. Historical records of elemental concentration ratios ((a) Si/Al, (b) nssK/Al, (c) Fe/Al, (d) nssMg/Al, (e) nssCa/Al, and (f) average volume of dust particles in the EGRIP ice core.

Major Comments:

The authors mostly compare their results with an ice core from northeast Greenland (sigma-d) for which similar data are available. However, the comparison to central Greenland, east Greenland (Renland) and southeast Greenland (Dye-3) should be included in the discussion. In particular, the comparison with NGRIP should be made, as the claim that EGRIP represents Eastern Greenland and NGRIP central Greenland is a bit shaky, considering both sites are at similar altitudes and quite close to each other.

We obtained mineral composition data from Central Greenland ice cores (GRIP and GISP2; e.g., Maggi, 1997; Svensson et al., 2000; Újvári et al., 2022) and compared the results with the dust from the EGRIP ice core (Fig. Reply 1). The lower quartz abundance may indicate that the EGRIP dust was contributed not only from Asia but also from other source areas. Additionally, we found one dataset from east and southeast Greenland (Renland and Dye-3; Bory et al., 2003), providing information on clay mineral composition (Fig. Reply 2). The clay mineral composition of the EGRIP ice core dust was not similar to that of the Renland and Dye-3 ice core dust. However, we cannot discuss about the similarity of clay mineralogy between the ice cores based on only one dataset.

The authors group Europe and NorthEast Asia, as well as Africa and SouthEast Asia into single potential source areas in their analysis. Considering the long debate about Asian, European and African dust sources for Greenland, these should probably be split into four, unless the authors can justify their choice.

As shown in first response, we recalculated the back-trajectories as follows. Potential dust source (land) areas are divided into eight regions following referee #1's suggestion; the Greenland Ice Sheet (GrIS), the Greenland coast (GrC), North America (NA), Europe (EU), Russia (RUS), Central Asia (CA), Southeast Asia (SA), Middle East (ME), and Africa (AF) (Fig. Reply 5). Considering the dust sources, the ocean and GrIS were excluded from the calculation.

The authors talk about trends in the data in various sections of the manuscript, in particular comparing the last 20 years with the mid-section of the core. In particular, the authors imply that the recent warming has been responsible for various changes in dust mineralogy and concentrations. But looking at the complete record, these look more like multidecadal oscillations to me and should not be described as trends. Although recent warming in Greenland may have been responsible for some of the observed changes, such a hypothesis has to be put in context with the complete oscillations shown in the records. The authors briefly mention NAO at some point, but the link between their record and various Atlantic and Pacific oscillations should be discussed in much more details.

We will revise the sentences to describe there is a local supply albeit small as suggested by another referee. We will carry out analyses related to Atlantic and Pacific oscillations as described above.

[Figure]

Figure reply5 (revised from fig. 2): Map showing (a) location of EGRIP and SIGMA-D ice core sites in Greenland and five regions used for calculating regional contribution (GrIS: Greenland Ice Sheet, grey; GrC: Greenland coast, purple; NA: North America, orange; EU: Europe, blue; RUS: Russia, light blue; CA: Central Asia, yellow; SA: Southeast Asia, green; ME: Middle East, red; and AF: Africa, brown) and probability distribution for air mass at (b) EGRIP and (c) SIGMA-D sites from 7-day three-dimensional back trajectory analysis from 1958 to 2014.

Minor Comments:

Line 18: Abstract could benefit from a more general introductory phrase at the beginning.

We will add a more general introductory phrase at the beginning of the abstract as suggested by the referee.

Line 33-34: Since Greenland is an island, all air masses must come from a coast. Be more precise.

We will delete sentences describing air mass contribution from Greenland coast as suggested by another referee.

Line 38: 100,000 years seems a bit short for the geological timescale, although I am not a geologist and may be wrong. Maybe Milankovitch timescale?

We agree with your suggestion and will revise the sentences to read "During the glacial-interglacial climate changes (e.g., from the Eemian to the Holocene), ice-core dust records…".

Line 42: "ice-core dust shows…". Also this is only shown for Central Greenland, not the whole of Greenland.

We will revise the sentences to read "In the 20th century, the Central Greenland ice-core dust showed…" in the revised manuscript.

Line 44-45: This is not at all the message of Svensson et al., 2000. Generally, I very much doubt that the seasonal variability in dust advection to Greenland is due to climate change…

We will revise the sentences to read "…climate change, environmental changes in dust sources such as their extent and aridity, and atmospheric transport"

Line 46: Not "predict". "estimate" maybe.

We will change the words "predict" to "estimate" in the revised manuscript.

Line 47: "partly responsible". Grain size and partial melt is very important as well for albedo.

We will add the "partly" before "responsible" as suggested by referee.

Line 53-58: The message of these phrases is unclear. Are you suggesting to collect dust from outcropped ice in the ablation zone to measure old dust? Or just dust in fresh snow on the surface? Then why talk about the movement of ice and dust through the ice sheet?

Nagatsuka et al. (2016) reported that englacial dust appear to be important for forming the dark ice surface of a glacier in northwestern Greenland. The dust was deposited from the atmosphere in the past in the upstream, traveled through the ice sheet, outcropped again in the ablation zone,

and formed cryoconite. We assume that the dust was likely deposited widely on the ice sheet in the past and thus preserved in ice cores. We will add these descriptions in the revised manuscript.

Line 60: Ujvari et al. used Hf not Pb.

We will correct "Hf" to "Pb" in the revised manuscript.

Line 75: Can you give some references to support that hypothesis?

We will add some references (e.g. Bory et al 2003; Kjær et al 2022) to support the hypothesis.

Line 117: Is the Beckman CC located in a normal laboratory or a clean room or a laminar flow bench? What kind of aperture tube was used?

Beckman CC is located in clean room in a Class 10000 clean room and we use $30\,\mu$ m aperture tube.

Line 145, Table 1: Why is South America included as a possible source for Type A particles? I'm not saying it's wrong (although I do doubt it), but I wonder why it was included in the list.

It was a mistake. We will delete South America from Table 1.

Lines 171-174: Snow cover fractions vary substantially from one model to another. Please provide and uncertainty estimate due to the choice of the model.

Following the suggestion, we will analyze snow cover fractions using LS3MIP multi-model results to evaluate model uncertainty.

Lines 209-216: What do the numerical ranges indicate? 1-sigma range? If so how were the mean and standard deviations calculated?

This range represents the minimum to maximum values of particle diameter. We will add a description explaining about it in the manuscript.

References:

Bory, A. J.-M., Biscaye, P. E., Piotrowski, A. M., and Steffensen, J. P.: Regional variability of ice core dust composition and provenance in Greenland, Geochem. Geophys. Geosyst., 4, 1107, https://doi.org/10.1029/2003GC000627, 2003.

Formenti, P., Schütz, L., Balkanski, Y., Desboeufs, K., Ebert, M., Kandler, K., Petzold, A., Scheuvens, D., Weinbruch, S., and Zhang, D.: Recent progress in understanding physical and

chemical properties of African and Asian mineral dust, Atmos. Chem. Phys., 11, 8231–8256, https://doi.org/10.5194/acp-11-8231-2011, 2011.

Han, C., Hur, S. D., Han, Y., Lee, K. Hong, S., Erhardt, T., Fischer, H., Svensson, A. M., Steffensen, J. P., Vallelonga, P.: High-resolution isotopic evidence for a potential Saharan provenance of Greenland glacial dust, Sci. Rep., 8, 15582, | https://doi.org/10.1038/s41598-018-33859-0 3, 2018.

Kjær, H. A., Zens, P., Black, S., Lund, K. H., Svensson, A., and Vallelonga, P.: Canadian forest fires, Icelandic volcanoes and increased local dust observed in six shallow Greenland firn cores, Clim. Past, 18, 2211–2230, https://doi.org/10.5194/cp-18-2211-2022, 2022.

Maggi, V.: Mineralogy of atmospheric microparticles deposited along the Greenland Ice Core Project ice core, J. Geophys. Res., 102, 26725–26734, https://doi.org/10.1029/97JC00613, 1997.

Simonsen, M. F., Baccolo, G., Blunier, T., Borunda, A., Delmonte, B., Frei, R., Goldstein, S., Grinsted, A., Kjær, H. A., Sowers, T., Svensson, A., Vinther, B., Vladimirova, D., Winckler, G., Winstrup, M., and Vallelonga, P.: East Greenland ice core dust record reveals timing of Greenland ice sheet advance and retreat, Nat. Commun., 10, 4494, https://doi.org/10.1038/s41467-019-12546-2, 2019.

Svensson, A., Biscaye, P. E., and Grousset, F. E.: Characterization of late glacial continental dust in the greenland ice core project ice core, J. Geophys. Res., 105, 4637–4656, https://doi.org/10.1029/1999JD901093, 2000.

Újvári, G., Klötzli, U., Stevens, T., Svensson, A., Ludwig, P., Vennemann, T., Gier, S., Horschinegg, M., Palcsu, L., Hippler, D. Kovács, J., Di Biagio, C. and Formenti, P.: Greenland ice core record of last glacial dust sources and atmospheric circulation. J. Geophys. Res. Atmo.s, 127, e2022JD036597, https://doi.org/10.1029/2022JD036597, 2022.

Zhang, Y., Mahowald, N., Scanza, R. A., Journet, E., Desboeufs, K., Albani, S., Kok, J. F., Zhuang, G., Chen, Y., Cohen, D. D., Paytan, A., Patey, M. D., Achterberg, E. P., Engelbrecht, J. P., and Fomba, K. W.: Modeling the global emission, transport and deposition of trace elements associated with mineral dust, Biogeosciences, 12, 5771–5792, https://doi.org/10.5194/bg-12-5771-2015, 2015.

---

## Author Comment (AC4)

*Reply to Referee#2*

December 19, 2023

Dear Referee #2,

We greatly appreciate a number of valuable comments on our manuscript entitled "Regional variations in mineralogy of dust in ice cores obtained from northeastern and northwestern Greenland over the past 100 years" by Nagatsuka et al. submitted to the journal Climate of the Past. Please see enclosed our responses (**blue text**) to each of your comments (**black text**) describing on the following pages.

Best regards,

Naoko Nagatsuka and co-authors
National Insitute of Polar Reserach
E-mail: nagatsuka.naoko@nipr.ac.jp

General statements

This paper documents analyses of dust particles in the EGRIP Greenland ice core. The analyses consist of size information, and a set of SEM observations that have allowed mineralogical classification to be made. The work covers only the last century in 10 decade-long bins. The data are certainly worthwhile to make available, and a significant amount of work has gone into the SEM study. However it must be said that the new insights gained from this study are quite minor. I appreciate the authors' point that isotopic analyses would be demanding in terms of sample size (though not with modern instruments quite as demanding as implied); however the coarse mineralogical separation here is simply not capable of defining source areas. It might (as in the previous work from SIGMA-D) be capable of defining the appearance of local sources but in this work no evidence of any local material is presented. Additionally the time series all appear flat (within statistical variation) meaning that there is no story about changing sources here. The back trajectory work is used in a way I think is inappropriate to try to define source areas (I will elaborate later). The work about volcanic material presents nothing new – we would not expect to see tephra from most eruptions, and indeed we don't. In summary, there are data here that are worthwhile to make available, but there is no scientific story, and the data are not capable of defining source areas. It is therefore for the editor to decide: the authors could be encouraged to strip the paper down and correct/remove unsupported statements so that a correct but unexciting paper appears in CP: this would be a major revision as there are some issues that are conceptually wrong (especially regarding back trajectories) at present. Or they could recommend submission of a stripped down version to a journal such as ESSD that takes datasets without expecting too much in the way of interpretation.

We would like to thank you for all the valuable comments. We agree with your suggestions that no firm evidence of local material is presented in this study and that coarse mineralogical separation here is not enough to define source areas. Thus, we will remove Fig.11 and revise the sentences to describe there is a local supply albeit small as you suggested. We also agree that accurately 'identifying' the sources of the ice core dust is challenging based solely on trajectory results. However, our results clearly demonstrate the differences in dust sources between the EGRIP and SIGMA-D ice cores based on mineral composition. We will revise the manuscript to emphasize the distinctions between the two ice core sites and 'discuss' potential dust source areas. Consequently, one of the possible sources of the EGRIP ice core dust is likely to be Asian deserts as the other Greenland ice core dust from glacial periods and the time series of the mineral composition did not change significantly over the last 100 years. These results may not sound exciting or new. But we argue that these results are new and important since no previous study has confirmed it. Although previous studies revealed mineral dust sources of the Greenland ice cores, most of the dust were obtained from the last glacial period when the dust concentrations

were high. Furthermore, we also found the new results indicating that the source areas have changed since 1970-1980, which is new insight that could improve the understanding of Greenland ice core dust. According to the suggestions from you and other referees, we will add the results of new analyses and revise our manuscript as below:

Introduction and conclusion:

First, we will clarify and emphasize new and important findings in our study, especially in introduction and conclusion sections. Our SEM-EDS analysis show the following two new results: This study is the first to demonstrate (1) continuous records of a northeast Greenland ice core dust (EGRIP) composition with a high temporal resolution during the last 100 years and (2) their significant differences from northwestern Greenland.

Although previous studies revealed mineral dust sources of the Greenland ice cores, most of the dust were obtained from the last glacial period when the dust concentrations were high. Some studies have analyzed the ice core minerals during the Holocene (8.2-11.6 ka : Han et al., 2018, 4-7 ka: Simonsen et al., 2019, 1997-2001 A.D : Bory et al., 2003a), a period of low dust concentration, however, they needed to concentrate decades to thousands of years of ice for each sample. Thus, our results can provide new and important insight to understand sources and transportation processes of ice core dust as well as climate and environmental change in recent years when global warming is progressing.

Results and Discussion:

We agree that it is difficult to "identity" the sources of the ice core dust accurately only based on our mineralogical and trajectory results as suggested by referee #2. However, our results can clearly show the differences in the dust sources between the EGRIP and SIGMA-D ice cores based on the mineral composition. Thus, we will revise the manuscript to emphasize the differences in the two ice core sites and "discuss" possible dust source areas. We will also add some new results of the mineral composition and elemental concentration of the EGRIP ice core dust obtained from the SEM-EDS and ICP-MS analyses to discuss the possible sources and their temporal variations (Figs. Reply. 1 - 4). The details are as follows.

The EGRIP dust showed mineral composition characterized in Asian-sourced dust as described in line 305-307. Thus, Asian desert can be one of possible sources of the dust at EGRIP as the other Greenland ice core sites. Comparing the mineral composition, however, the EGRIP ice core dust showed significantly lower quartz content and higher illite/micas/chlorite contents than the GRIP ice core dust originated from Chinese deserts (Fig. Reply 1 and 2). We cannot compare the EGRIP mineral composition records directly with those of previous studies because there have been no continuous records of dust composition during the last 100 years. Moreover, the periods

analyzed (modern vs mostly glacial), analytical methods (X-ray diffraction analysis (XRD) vs SEM-EDS) and mineral identification methods of the ice core dust were different (XRD used in previous studies cannot separate micas from illite, whereas the elemental peak intensity ratio sorting scheme used in our study cannot separate micas from chlorite completely. We identified them as mixed layer of micas/chlorite). However, the lower quartz abundance may indicate that the EGRIP dust were contributed not only from Asia, but also from other source areas. Thus, the EGRIP ice core dust was originated from multiple sources. Air-mass trajectory results suggests that Northern Eurasia and North America are the best candidate of the other possible sources.

[Figure]

Figure Reply 1. Mineral composition of the EGRIP (this study) and GRIP ice cores (Svensson et al., 2000)

[Figure]

Figure Reply 2. Clay mineral composition of the EGRIP (this study) and Hans Tausen, NGRIP, GRIP, Renland, Site A, and Dye-3 ice cores (Bory et al., 2003).

Although mineral composition records indicates that the mineral sources of the EGRIP dust have not changed dramatically over the past 100 years as described in Line 26, the ternary clay mineralogy diagram (Fig. 10) and the elemental concentration ratios of the EGRIP ice core dust analyzed by the SEM-EDS and ICP-MS, respectively, suggest that there is slight difference in the dust sources in different periods. In ternary clay mineralogy diagram, the dust from 1910 to 1970 were plotted close to glacial dust from the GRIP ice core as well as the Asian dust, whereas the dust from 1970 to 2013 were not. Furthermore, the samples from 1980 to 2013 showed the steeper slope in scatter plots of elemental concentrations analyzed by ICP-MS (Fig. Reply 3). These results indicate that the contribution rate from the possible sources has changed since 1970-1980, which is new insight that could improve the understanding of Greenland ice core dust.

One of the possible causes of this change is decreasing Asian dust supply to the EGRIP. Although the number of the particles may not be sufficient to show accurate percentages, the illite content of the EGRIP showed decreasing trend since 1980 that is similar to that for dust occurrence in Asian deserts (e.g. Zhang et al., 2020). This is likely due to global warming that could have led to a decrease in atmospheric temperature gradients and decline in wind speed and then decreasing dust storm frequency/intensity. We also compared the nss Ca/Al ratios with average volume of dust particles calculated from volume concentration divided by number concentration of the EGRIP ice core, which can reflect the contribution of evaporite minerals that are abundant in deserts (Formenti et al., 2011) and the distance from the source areas, respectively (Fig. Reply 4). The nss Ca/Al ratios were slightly lower but average volume of dust particles were higher from 1970 to 1990. These results indicate that the contribution from distant deserts likely decreased during the period, which are consistent with the hypothesis that the dust contribution has changed since 1970-1980.

The higher Ca/Al ratios and lower average volume of dust particles of the EGRIP dust after 1990 imply the possibilities that the dust has originated from distant sources with higher salinity environments in this period. Previous studies have revealed that there are large variations in the Ca/Al ratios among Asian deserts and that the Taklamakan deserts has higher values than the other deserts (e.g. Formenti et al., 2011). The EGRIP ice core dust might be derived from such sources.

We will add some sentences in the manuscript to describe these points.

[Figure]

Figure Reply 3. Elemental concentration ratios of microparticles in the EGRIP ice core anayzed by ICP-MS.

[Figure]

Figure Reply 4. Historical records of elemental concentration ratios ((a) Si/Al, (b) nssK/Al, (c) Fe/Al, (d) nssMg/Al, (e) nssCa/Al, and (f) average volume of dust particles in the EGRIP ice core.

Detailed comments

Abstract. There are a couple of sentences I don't think are supported by the data and text. On lines 26-27 (local dust), I don't think the case is made at all for a local dust source.

We will revise the abstract in a way that is consistent with our results and delete the sentence "The subtle variation in the EGRIP ice-core mineral composition is likely due to a minor contribution of local dust."

On lines 31-33, the back trajectories really don't make this case as presented.

As described in the response to general statement, we will revise the manuscript to emphasize the difference in dust sources between the EGRIP and SIGMA-D and tone down our description about identifying dust sources. Thus, we can use our trajectory results to indicate the significant differences in the dust contribution from possible sources between the two ice core sites in the revised manuscript although we cannot use to accurate quantitative evaluation.

And the last sentence is contradictory as written.

We will delete the last sentences.

Section 2.3: please be more precise with this explanation. As written, my reading is that you collected a total of 11 filters, and analysed 200 particles on each. However was each sample an average of all the vials from the 10 year interval or is it a spot measurement from one or more vials within the section? This is crucial to whether the samples are representative.

Many thanks for the suggestion. Each sample was an average of the 10 year-ice samples. For the SEM analysis, the ice core samples were first melted and collected in the glass vial for every 10-year and well stirred. Next, 500μL of each sample was extracted from the vial and was filtered through a polycarbonate membrane filter (So we made 11 filters). Assuming that the dust samples were distributed uniformly on a filter, we cut a filer into quarters and observed a total of 200 randomly selected particles from it. To eliminate bias for choice of mineral dust as much as possible, we observed particles in various position on a 1/4 filter over a couple of days. We will add the sentences to describe above in the Section 2.3.

And if you only analysed 200 particles per filter, then how are you later giving mineralogies to 0.1% accuracy (Table 3)? If this is really what was done (I hope not) then most of the differences between time periods would be completely statistically insignificant, so it's really important to understand this.

Fig 6. I again emphasise that if each decade is really only represented by a count of the types of 200 particles then most differences are likely to be statistically insignificant so again please clarify

in methods, and if I have understood correctly please don't overinterpret counts that have large uncertainties on them.

Section 4.1 eg line 263. Again I doubt the significance of any variability here given the number of particles of each sort counted.

We agree on the referee's comments that 200 particles are likely insufficient to show accurate proportion of each mineral, and thus we will not show decimal point of the data. However, we think our results can clearly show that the size and composition and their variations of the EGRIP ice core dust differ substantially from the SIGMA-D ice core dust analyzed in the same way as this study (150 particles in every 5 years for the SIGMA-D) over the past 100 years. Thus, again, we will revise our manuscript to emphasize the differences in the sources between the EGRIP and SIGMA-D ice core dust. We will also analyze some remaining filters and confirm that their mineral composition will be the same as this study.

Back trajectories: Starting with lines 153-5. I don't understand what you are telling us about wet deposition here? Standard HYSPILT trajectories are simply that – trajectories of air masses, which take no account of the contents of the air and therefore take no account of what is lost en route or how it is deposited in Greenland. I am therefore not clear what you mean about wet deposition and precipitation. Please explain but for now I will assume you present simple back trajectories.

"was considered" (L155) may be an insufficient description. When we calculated the probability, we accumulated number of air mass within a given 1deg x 1deg grid, and then obtained the percentage against the total count at the grid. To consider the precipitation effect, we used the precipitation amount when the air mass arrived at the site. By this consideration, we can assume that the air mass without precipitation (0 mm) doesn't contribute to the probability. We will change the description L154-155 as: (Iizuka et al., 2018; Parvin et al., 2019), for which the probability was weighted by the daily precipitation at the date when the air mass arrived at the ice core site (Nagatsuka et al., 2021).

The next issue is that you describe launching trajectories from 4 different altitudes, but you don't say which is shown in your figures. I assume it's all 4 mixed together but this is an odd thing to do without first discussing what are the differences for different altitudes of launch.

We show the probability distributions of the 7-day back trajectories for the four different altitudes of the initial air mass (Fig. Reply 5). The figure shows the relatively higher contribution from distant regions (North America and Northern Eurasia) and lower contribution from Greenland coast at high altitudes. The differences in such contributions between the altitudes are smaller in the EGRIP, but larger in the SIGMA-D (the contribution from North America increase

significantly at high latitudes). However, we don't think that it's necessary to change Fig. 2 to Fig. Reply 5 in the revised manuscript by the following three reasons: (1) the discussion about dust source areas will be mainly based on the SEM-EDS results as described in the response to general comments, and not quite depend on the trajectory results. (2) Fig. 2 can clearly show the differences in the airmass transportation pathways between the EGRIP and SIGMA-D. (3) previous studies have not shown nor discussed the results at different altitudes.

[Figure]

Figure Reply 5. Probability distribution for air mass at EGRIP and SIGMA-D sites in four different altitudes (50, 500, 1000, 1500m) from 7-day three-dimensional back trajectory analysis from 1958 to 2014.

Then Figure 2 is useless to the reader as the trajectory densities shown are nearly all over the ocean which cannot be a source of dust. You need to treat the trajectories in a different way: so we can judge where they might pick up dust. But there are a number of subtleties that are not well treated here. Firstly, of course every trajectory passes over the Greenland coast where there might be rock. But what matters is (a) how long the air spent over a source and (b) whether it was at low altitude where it could pick up fresh material (bearing in mind that as Schupbach discussed there could be places where air is lofted to altitude from the surface but at scales that Hysplit doesn't capture). The reason I mention this in respect of Fig 2 is that the Figure gives the impression of a high input from the Greenland coast, but the reality is that the air probably only spends an hour or less over the thin coastal strip of sediment/soil, and probably not at ground level (as the air has to have reached high altitude by the time it reaches EGRIP). The figure is therefore misleading about the potential influence of local material, and completely unsuited (because the trajectories are too short) to showing which distant continent could have contributed.

We would appreciate it if it was clear what "a different way" was. About your "matters", we cannot describe (a) how long the air mass stayed over a grid as other similar studies didn't. But the altitude for summing up air mass probability was constrained lower than 1500 a.g.l. by assuming the mixing layer with the description "within this altitude range" (L153). For clarity, we will add "(< 1500 m a.g.l.)" after "within this altitude range" (L153).

Page 7 re dating. This is OK but I'd like to have seen your assignment of the ages near the bottom of your section confirmed by deeper volcanic matches (ie ones below the ice you used). Motjabavi et al 2020 have presented a chronology for EGRIP, so it would be helpful if you compared your chronology to theirs.

Many thanks for the suggestion. We will describe deeper volcanic events to confirm the ages of our bottom section of the ice core.

Line 215. "The mode values showed an increasing trend over the 100-year period except for the 1910–1920 sample". This is not an acceptable statement. With that first sample, there is no trend, and even without it, I doubt the trend is statistically significant.

We will revise the sentences that "The mode diameter showed an almost similar values (0.37-0.48µm) during the periods, except for a large value of the 1910–1920 sample (0.57µm,), but was slightly higher of the 1960-2013 samples than the 1920-1960 samples. "

Fig 7 should be removed. It simply doesn't represent what you say for the reasons I outlined above. These are not the "contributions of air masses" unless you weight them by the length of time they spend over a location (in fact I am very unclear what is plotted here anyway but it certainly isn't what it says). Fig 8 is slightly more helpful and could possibly be used to interpret your data if we also had information about altitudes (where did trajectories last intersect with the ground?).

We will remove Fig 7. About Fig. 8, we clarify the altitude we dealt with in the reply above addressed. We show trajectory results for the four different altitudes in Fig. Reply 5.

Line 282. "The morphological properties of the EGRIP ice-core dust also suggest a small supply of minerals from local source areas". This is not correctly written. It suggests evidence that there is a local supply albeit small. What you actually have is no evidence for a local supply. Please reword.

We will reword the sentences as suggested by Referee #2.

Page 15/16. While I agree that prior data suggest an East Asian source I cannot agree that you have added new evidence for that. As your Table 1 shows, 4 of your dust types could originate from East Asia but none of them uniquely so. Indeed both Asia and North America are mentioned for types A, B, C. This method is simply incapable to differentiate the long range sources.

Although the EGRIP ice core dust was likely originated from multiple distant sources (Asia, North America and Northern Eurasia), it is impossible to identify contribution rate from each region based on the mineral composition as suggested by the referee because of the similar characteristics of the composition. However, the ternary clay mineralogy diagram (Fig. 10) can clearly distinguish Asia and North America and the EGRIP ice core dust from 1910 to 1970 were plotted close to glacial dust from the GRIP ice core as well as the Asian dust. Furthermore, the EGRIP ice core dust showed significantly lower kaolinite contents than the SIGMA-D dust that was likely derived from northern Canada, which is also supported by trajectory results (Fig. 8). Thus, we think that the contribution from North America to the EGRIP was smaller than the SIGMA-D, and that from Asia was likely larger until 1970-1980.

The discussion of trajectories on page 16 is also incapable of differentiating sources as discussed above (and also for the reasons given by Schupbach).

We agree that the back-trajectory analysis is incapable of identifying dust transport from Asia to Greenland as suggested by Schüpbach et al. (2018) and thus is not used to show the accurate contribution rate from each possible source area. However, our trajectory results show the clear difference between the EGRIP and SIGMA-D sites.

Page 17. I simply can't see the possible changes (more after 1980) in proportion of medium sized particles in Figure 11. Please either make a clear statistical case or remove this implication. Additionally changes in the proportion of 2 micron particles have not been shown to indicate a local source, but more likely a change in transport strength (if significant).

We agree with your suggestions and remove this implication and Figure 11.

Section 4.3 adds nothing and should be removed.

We will remove Section 4.3 as suggested by referee.

The conclusions need to be changed to reflect the changes in the text, eg lines 432-4 are not shown in the data.

We will revise the conclusion to reflect the discussion.

References:

Bory, A. J.-M., Biscaye, P. E., Piotrowski, A. M., and Steffensen, J. P.: Regional variability of ice core dust composition and provenance in Greenland, Geochem. Geophys. Geosyst., 4, 1107, https://doi.org/10.1029/2003GC000627, 2003.

Formenti, P., Schütz, L., Balkanski, Y., Desboeufs, K., Ebert, M., Kandler, K., Petzold, A., Scheuvens, D., Weinbruch, S., and Zhang, D.: Recent progress in understanding physical and chemical properties of African and Asian mineral dust, Atmos. Chem. Phys., 11, 8231–8256, https://doi.org/10.5194/acp-11-8231-2011, 2011.

Han, C., Hur, S. D., Han, Y., Lee, K. Hong, S., Erhardt, T., Fischer, H., Svensson, A. M., Steffensen, J. P., Vallelonga, P.: High-resolution isotopic evidence for a potential Saharan provenance of Greenland glacial dust, Sci. Rep., 8, 15582, | https://doi.org/10.1038/s41598-018-33859-0 3, 2018.

Iizuka, Y., Uemura, R., Fujita, K., Hattori, S., Seki, O., Miyamoto, C., Suzuki, T., Yoshida, N., Motoyama, H. and Matoba, S.: A 60 year record of atmospheric aerosol depositions preserved in a high-accumulation dome ice core, Southeast Greenland, J. Geophys. Res., 123, 574–589. https://doi.org/10.1002/2017JD026733, 2018.

Maggi, V.: Mineralogy of atmospheric microparticles deposited along the Greenland Ice Core Project ice core, J. Geophys. Res., 102, 26725–26734, https://doi.org/10.1029/97JC00613, 1997.

Nagatsuka, N., Goto-Azuma, K., Tsushima, A., Fujita, K., Matoba, S., Onuma, Y., Dallmayr, R., Kadota, M., Hirabayashi, M., Ogata, J., Ogawa-Tsukagawa, Y., Kitamura, K., Minowa, M., Komuro, Y., Motoyama, H., and Aoki, T.: Variations in mineralogy of dust in an ice core

obtained from northwestern Greenland over the past 100 years, Clim. Past., 17, 1341–1362, https://doi.org/10.5194/cp-17-1341-2021, 2021.

Parvin, F., Seki, O. Fujita, K., Iizuka, Y., Matoba, S. and Ando, T.: Assessment for paleoclimatic utility of biomass burning tracers in SE-Dome ice core, Greenland, Atmos. Environ., 196, 86–94, https://doi.org/10.1016/j.atmosenv.2018.10.012, 2018.

Schüpbach, S., Fischer, H., Bigler, M., Erhardt, T., Gfeller, G., Leuenberger, D., Mini, O., Mulvaney, R., Abram, N. J., Fleet, L., Frey, M. M., Thomas, E., Svensson, A., Dahl-Jensen, D., Kettner, E., Kjaer, H., Seierstad, I., Steffensen, J. P., Rasmussen, S. O., Vallelonga, P., Winstrup, M., Wegner, A., Twarloh, B., Wolff, K., Schmidt, K., Goto-Azuma, K., Kuramoto, T., Hirabayashi, M., Uetake, J., Zheng, J., Bourgeois, J., Fisher, D., Zhiheng, D., Xiao, C., Legrand, M., Spolaor, A., Gabrieli, J., Barbante, C., Kang, J.-H., Hur, S. D., Hong, S. B., Hwang, H. J., Hong, S., Hansson, M., Iizuka, Y., Oyabu, I., Muscheler, R., Adolphi, F., Maselli, O., McConnell J., and Wolff, E. W.: Greenland records of aerosol source and atmospheric lifetime changes from the Eemian to the Holocene, Nat. Commun., 9, 1476, https://doi.org/10.1038/s41467-018-03924-3, 2018.

Simonsen, M. F., Baccolo, G., Blunier, T., Borunda, A., Delmonte, B., Frei, R., Goldstein, S., Grinsted, A., Kjær, H. A., Sowers, T., Svensson, A., Vinther, B., Vladimirova, D., Winckler, G., Winstrup, M., and Vallelonga, P.: East Greenland ice core dust record reveals timing of Greenland ice sheet advance and retreat, Nat. Commun., 10, 4494, https://doi.org/10.1038/s41467-019-12546-2, 2019.

Svensson, A., Biscaye, P. E., and Grousset, F. E.: Characterization of late glacial continental dust in the greenland ice core project ice core, J. Geophys. Res., 105, 4637–4656, https://doi.org/10.1029/1999JD901093, 2000.

Újvári, G., Klötzli, U., Stevens, T., Svensson, A., Ludwig, P., Vennemann, T., Gier, S., Horschinegg, M., Palcsu, L., Hippler, D. Kovács, J., Di Biagio, C. and Formenti, P.: Greenland ice core record of last glacial dust sources and atmospheric circulation. J. Geophys. Res. Atmo.s, 127, e2022JD036597, https://doi.org/10.1029/2022JD036597, 2022.

Zhang, Y., Mahowald, N., Scanza, R. A., Journet, E., Desboeufs, K., Albani, S., Kok, J. F., Zhuang, G., Chen, Y., Cohen, D. D., Paytan, A., Patey, M. D., Achterberg, E. P., Engelbrecht, J. P., and Fomba, K. W.: Modeling the global emission, transport and deposition of trace elements associated with mineral dust, Biogeosciences, 12, 5771–5792, https://doi.org/10.5194/bg-12-5771-2015, 2015.